METHODS AND RESOURCES

# Synaptic and dendritic architecture of different types of hippocampal somatostatin interneurons

**Virág Takács**[1◉], **Zsuzsanna Bardóczi**[1◉], **Áron Orosz**[1,2◉], **Abel Major**[1◉], **Luca Tar**[1,3◉], **Péter Berki**[1,2], **Péter Papp**[1], **Márton I. Mayer**[1,2], **Hunor Sebők**[1], **Luca Zsolt**[1], **Katalin E. Sos**[1,2], **Szabolcs Káli**[1], **Tamás F. Freund**[1], **Gábor Nyiri** [1]*

1 Laboratory of Cerebral Cortex Research, HUN-REN Institute of Experimental Medicine, Budapest, Hungary, 2 János Szentágothai Doctoral School of Neurosciences, Semmelweis University, Budapest, Hungary, 3 Roska Tamás Doctoral School of Sciences and Technology, Pázmány Péter Catholic University, Budapest, Hungary

◉ These authors contributed equally to this work.
* nyiri@koki.hu

**Data Availability Statement:** All data are fully available without restriction, all relevant data are included in the manuscript and its supporting

## Abstract

GABAergic inhibitory neurons fundamentally shape the activity and plasticity of cortical circuits. A major subset of these neurons contains somatostatin (SOM); these cells play crucial roles in neuroplasticity, learning, and memory in many brain areas including the hippocampus, and are implicated in several neuropsychiatric diseases and neurodegenerative disorders. Two main types of SOM-containing cells in area CA1 of the hippocampus are oriens-lacunosum-moleculare (OLM) cells and hippocampo-septal (HS) cells. These cell types show many similarities in their soma-dendritic architecture, but they have different axonal targets, display different activity patterns in vivo, and are thought to have distinct network functions. However, a complete understanding of the functional roles of these interneurons requires a precise description of their intrinsic computational properties and their synaptic interactions. In the current study we generated, analyzed, and make available several key data sets that enable a quantitative comparison of various anatomical and physiological properties of OLM and HS cells in mouse. The data set includes detailed scanning electron microscopy (SEM)-based 3D reconstructions of OLM and HS cells along with their excitatory and inhibitory synaptic inputs. Combining this core data set with other anatomical data, patch-clamp electrophysiology, and compartmental modeling, we examined the precise morphological structure, inputs, outputs, and basic physiological properties of these cells. Our results highlight key differences between OLM and HS cells, particularly regarding the density and distribution of their synaptic inputs and mitochondria. For example, we estimated that an OLM cell receives about 8,400, whereas an HS cell about 15,600 synaptic inputs, about 16% of which are GABAergic. Our data and models provide insight into the possible basis of the different functionality of OLM and HS cell types and supply essential information for more detailed functional models of these neurons and the hippocampal network.

information files, whereas the models and the source code for all the software developed for this study are available at https://zenodo.org/records/10580838. Models are also available at https://modeldb.science/2016137.

**Funding:** This work was supported by the Frontline Research Excellence Program of the Hungarian National Research, Development and Innovation Office (nkfih.gov.hu, NRDI Fund 133837), the Hungarian Brain Research Program NAP3.0 (agykutatas.hu, NAP2022-I-1/2022) and the European Union project RRF-2.3.1-21-2022-00011 within the framework of the Translational Neuroscience National Laboratory to G.N.; the European Union Human Brain Project (www.humanbrainproject.eu, SGA3, 945539) and the Hungarian Brain Research Program NAP2.0 (agykutatas.hu, 2017-1.2.1-NKP-2017-00002) to T.F.F. and G.N.; the European Union project RRF-2.3.1-21-2022-00004 within the framework of the Artificial Intelligence National Laboratory to S.K. and G.N.; the New National Excellence Program of the Ministry for Innovation and Technology and the Ministry of Human Capacities, (nkfih.gov.hu, ÚNKP-19-2-I-SE-36 and ÚNKP-18-2-I-SE-20 to A.M, ÚNKP-19-3-I-SE-9 to M.M, ÚNKP-23-2-II-DE-146 to H.S. and ÚNKP-23-3-I-SE-48 to Á.O.); and the National Academy of Scientist Education Program of the National Biomedical Foundation under the sponsorship of the Hungarian Ministry of Culture and Innovation to G. N. and H.S (www.edu-sci.org). The funders had no role in study design, data collection and analysis, decision to publish, or preparation of the manuscript.

**Competing interests:** The authors have declared that no competing interests exist.

**Abbreviations:** AAV, adeno-associated virus; BDA, biotinylated dextran amine; CES, Classical Evolutionary Strategy; Chrna2, nicotinic acetylcholine receptor alpha 2 subunit; DW, distilled water; eFEL, Electrophys Feature Extraction Library; EM, electron microscopic; HCS, high-conductance state; HS, hippocampo-septal; HSA, human serum albumin; MS, medial septum; OLM, oriens-lacunosum-moleculare; PBS, phosphate buffer solution; PC, pyramidal cell; PSP, postsynaptic potential; SEM, scanning electron microscopy; SOM, somatostatin; SWR, sharp wave-ripple; TBS, Tris-buffered saline; WT, wild-type.

## Introduction

A diverse population of GABAergic interneurons controls the signal integration and action potential firing of principal cells in the cerebral cortex [1–5]. A large group, 12% to 30% (depending on brain area) of these interneurons expresses the neuropeptide somatostatin (SOM) [6,7] and targets predominantly dendrites of local principal cells [8]. Somatostatin is a small peptide that localizes in dense-core axonal vesicles of SOM-positive neurons and acts as a co-transmitter modulating neuronal activity via both pre- and postsynaptic mechanisms [9]. Interestingly, SOM-positive interneurons play an indispensable role in memory processes and learning in all brain areas where they occur [10]. Two features of SOM-positive cells might explain this phenomenon: (i) with the location of their output inhibitory synapses on dendrites they are able to finely modulate the effect of glutamatergic inputs to principal cells [11]; and (ii) their input synapses show plasticity which can be important in long-lasting changes of the neuronal networks [12–18].

The dorsal hippocampus is essential for the formation of contextual memory and the hippocampal somatostatin-positive interneurons proved particularly important in the selection of engram cell assemblies responsible for the formation and recall of memories [8,19–24]. In the CA1 region of the dorsal hippocampus, somatostatin is expressed in several types of interneurons [8,25]. All of these innervate the dendrites of pyramidal cells (PCs) locally but their axonal arbors show preferences for different layers and co-align with different glutamatergic input pathways [8,26].

The most abundant type of locally projecting SOM-positive interneurons in CA1 are the oriens-lacunosum-moleculare (OLM) cells that were discovered by McBain and colleagues [27]. They have a horizontally oriented dendritic tree in str. oriens and a dense axonal arbor in str. lacunosum-moleculare. OLM cells receive most of their glutamatergic inputs from local CA1 PCs [28] and target the distal dendrites of the same population of cells [29,30]; therefore, they are ideally suited for feed-back regulation of the CA1 network [31]. Although in much smaller numbers, they also innervate interneurons, some of which interneurons target the proximal PC dendrites [29,32,33]. During contextual memory formation, CA1 PCs receive information about the context (multisensory information) from Schaffer collaterals of CA3 PCs to the proximal dendrites of CA1 PCs, and they integrate it with the unconditioned stimulus (e.g., an aversive stimulus) that is broadcasted by the temporo-ammonic pathway from the entorhinal cortical layer III PCs to the distal dendritic region of the CA1 PCs in str. lacunosum-moleculare. Because OLM cells target PC dendrites in str. lacunosum-moleculare, they can gate the information flow onto CA1 PC by directly excluding entorhinal inputs from a range of PCs. They can also facilitate the Schaffer collateral inputs via the disinhibition of proximal dendrites [33]. Furthermore, OLM cells can be activated by cholinergic and glutamatergic inputs from the medial septum (MS) [21,34] and inhibited by GABAergic inputs from the brainstem nucleus incertus [24], the latter of which proved a crucial component of memory formation in the hippocampus. These subcortical inputs can control which OLM cells can be activated during the memory formation and thus can indirectly determine the number of active CA1 PCs participating in the formation of a memory engram. Although the developmental origin of OLM cells was previously unclear, single-cell RNA sequencing data showed a very high homogeneity within the OLM cell population [35]. However, the expression of certain genes was detected only in subsets of OLM cells, which may suggest dynamically regulated temporal or permanent differences between OLM neurons [25,35].

Another group of CA1 somatostatin-positive neurons are primarily not local interneurons; instead, they send long-range axonal projections to other brain areas such as CA3, the dentate gyrus, the subiculum, the MS, and the medial entorhinal cortex [36–41]. Their main role may

be to temporally coordinate the activity of functionally related brain areas [37,42], which is important during memory processes. A large subgroup of these somatostatin-positive cells are the double-projecting cells which send long-range axons to both the MS and the subiculum and have sparse local axonal branches in CA1 str. oriens and radiatum [36,43–45]. Although in rats probably all septally projecting interneurons innervate the subiculum as well [36], no such data is available in the mouse yet. Here, we investigated these cells using retrograde labeling. Because we labeled them from the MS and could not examine their potential projections to other areas, we will use the classical terminology: we call them hippocampo-septal (HS) neurons in this study. Dendritic trees of HS cells are similar to those of OLM cells: These dendrites are horizontally distributed in str. oriens and are sparsely spiny [36,44,45]. In rats, they receive a very large number of glutamatergic inputs and a smaller amount of GABAergic innervation, a substantial proportion of which originates from GABAergic neurons of the MS [44]. Their local axons form output synapses predominantly with PC dendrites in adult rats [36,44], although HS cells were also found to target interneurons in in vitro slice preparations of juvenile rats [45].

Theta and gamma oscillations as well as sharp-wave-associated ripples accompany different stages of memory processing in the hippocampus [46–48]. OLM and HS cells have characteristically different firing patterns during sharp wave-ripples: OLM cells are silent or decrease their firing [49,50], whereas double projection (HS) cells discharge with high frequency during these events [36].

To understand the functional role of hippocampal interneurons and to build realistic models of the CA1 network, quantitative data are required about their morphological architecture and their connectivity [51]. The density and distribution of glutamatergic and GABAergic synaptic inputs, as well as the postsynaptic target selectivity are characteristically different features of different cell types [3,5,44,51–54]. Our knowledge of the anatomical parameters mentioned above is incomplete for many interneurons, especially in mice.

Here, we present the most complete reconstruction and analyses of the somato-dendritic and synaptic architecture of 2 classes of interneurons so far. We examined 2 types of SOM interneurons, the OLM and the HS cells because they play important roles in memory processes and the coordination of functionally related brain areas. These 2 cell types are located in the same layer, both express SOM and have similar dendritic morphology at the light microscopic level but they are functionally different. We tested whether their different functions are reflected in differences in their finer morphological details.

We cell-type-specifically labeled the 2 populations and reconstructed their whole dendritic trees at the light microscopic level. Then, we created 3D reconstructions of randomly selected dendritic ($n = 108$) and axonal ($n = 24$) segments of OLM and HS cells and measured several of their ultrastructural parameters using a scanning electron microscope. Semi-automatic acquisition of serial electron microscopic images enabled us to 3-dimensionally reconstruct thousands of synapses ($n = 5,844$) and correlate their anatomical parameters (size, linear density, surface density) to their location on the dendritic trees. We measured the electrophysiological properties of OLM and HS cells using in vitro hippocampal slices. To compare the signal propagation properties of OLM and HS cells, multi-compartmental passive and active models of OLM and HS cells were created by combining the data sets of the sampled dendritic segments and somata, the 3D morphology of cells, and the electrophysiological parameters.

Furthermore, we present all morphological and physiological raw data and calculated data in an easily accessible table format for future reference and modeling studies (Tables 1 and 2 and S1–S25 Data files).

## Results

### Identification of OLM and HS neurons

First, we labeled OLM and HS cells specifically. The workflow is illustrated in Fig 1. It has been repeatedly demonstrated that nicotinic acetylcholine receptor alpha 2 subunit (Chrna2) positive neurons in the dorsal hippocampus CA1 area, in stratum oriens are OLM cells [33,55,56]. Here, we labeled OLM cells specifically, using an injection of a Cre-dependent fluorescent reporter adeno-associated virus (AAV) into a Chrna2-Cre mouse. Additionally, because HS cells (by definition) project to the MS, they were labeled retrogradely from the MS. Depending on the type of data we collected, we used different retrograde tracer techniques including the injection of biotinylated dextran amine (BDA) or FluoSpheres into wild-type (WT) mice, or the injection of a Cre-dependent retrograde tracer into SOM-Cre mice (Fig 1A and 1B). After tracers labeled the target cell types, we specifically selected the type of tissue processing most suitable for collecting a given type of data set (Fig 1C, 1G, 1P and 1R).

### Identification of inputs

GABAergic synaptic surfaces were identified using the reliable preembedding immunogold labeling against gephyrin [57,58], a major postsynaptic protein in GABAergic synapses (Fig 2A and 2B). Based on data in the literature gephyrin labels only GABAergic synapses in cortical areas. Synapses that were not labeled for gephyrin were considered putative glutamatergic synapses, because (i) the ratio of false negative synapses are considered negligible because of the reliable gephyrin labeling that was tested on several sections of the same synapse; and (ii) although a negligible amount of non-GABAergic synapses may have been established by monoaminergic fibers, most of those fibers do not typically form synapses [59,60] in the CA1 stratum oriens. Therefore, we considered the labeled and unlabeled synapses putative GABAergic (p-GABAergic) and putative glutamatergic (p-glutamatergic) synapses, respectively.

### Reconstruction of unbiased anatomical parameters

To determine the synaptic, dendritic, and somatic parameters of OLM and HS cells that correspond to a live animal, we implemented a workflow that aims to be unbiased, representative and that compensates all distortions of the tissue processing procedures. Tissue shrinkage/dilatation was estimated during the whole process and corrected to collect data that represent in vivo parameters (see Methods).

OLM and HS cells were first reconstructed in light microscope and representative samples were taken from somata and dendritic segments that were further processed for scanning electron microscopy (SEM), after which they were reconstructed from 70 nm thick section series.

We collected the following data groups: high-resolution local morphological parameters of dendritic, somatic, and mitochondrial membranes; parameters of dendritic and somatic inhibitory and excitatory input synapses that OLM and HS cells receive; data for axon terminals that OLM and HS cells establish; light microscopic reconstructions of total dendritic arbors of OLM and HS cells; detailed in vitro physiological parameters. We are presenting all raw data and their statistical comparisons in supporting data files (S2–S10 Data and S19–S22 Data). In addition, we also used these data to build neuronal models of these cells to reveal their basic computational features and the functional consequences of their differences (Fig 1N, 1U and 1V).

### Morphologies of dendritic segments and branching points

Because the reconstructions of the total dendritic arbors of all cells examined here would have taken more than a decade, we took representative samples from these dendritic arbors, and

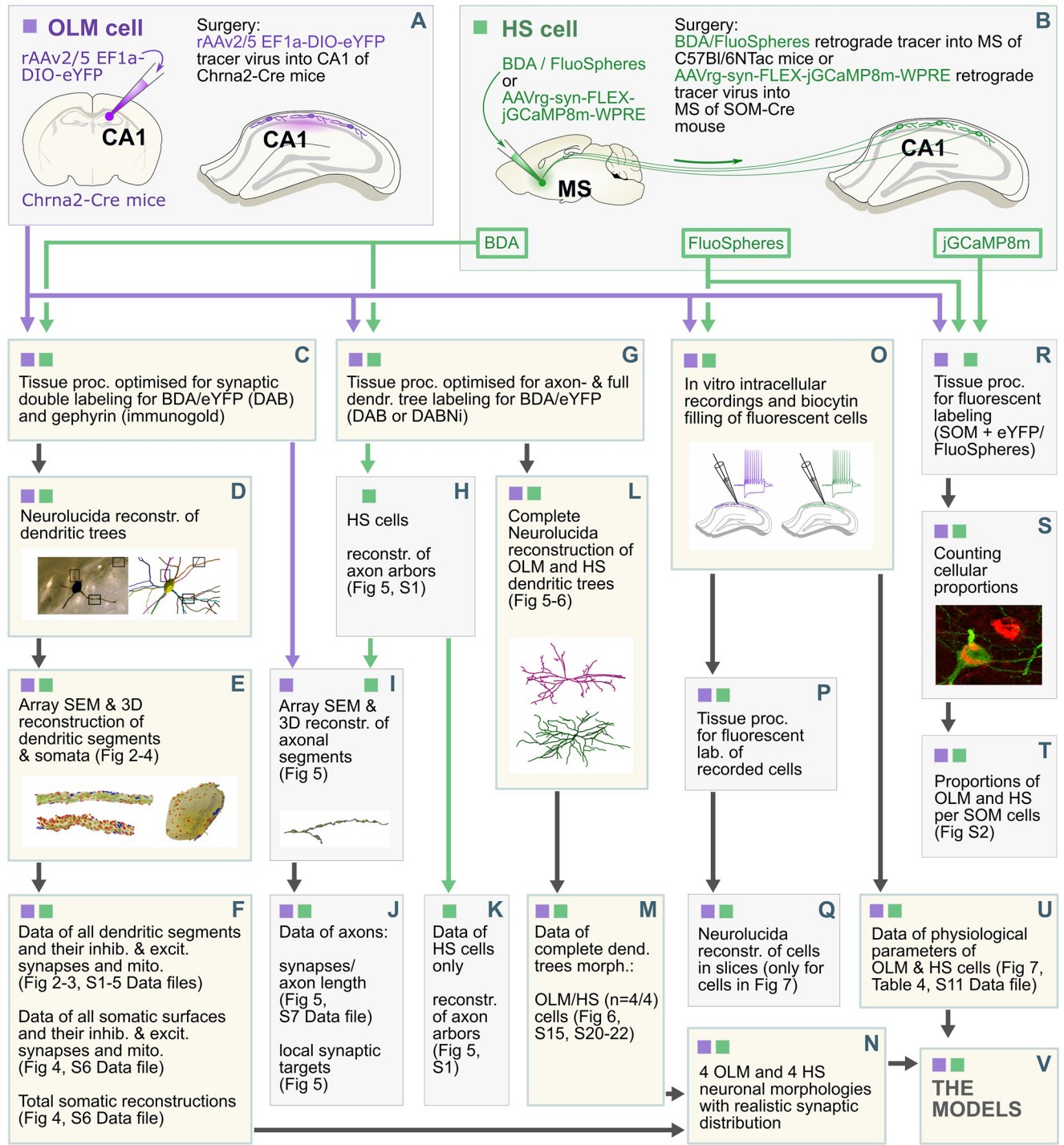

**Fig 1. Data collection workflow.** After cell type-specific labeling of OLM and HS cells using different tracing methods (A, B), samples went through different tissue processing protocols optimized for electron microscopic analysis of cells/whole cell reconstruction/colocalization experiments at light microscopic level or in vitro physiological recordings (C, G, O, P, R). Purple and green colors indicate OLM and HS cells, respectively. Black arrows indicate the workflow of sample preparation. Purple and green arrows indicate processes in which only OLM or HS cells were involved. Yellow boxes show the methods, work packages, which led to 3 major data sets in this study: the first data set contained all the representative dendrite type-specific data, including dendritic and somatic morphologies and synapses (F), second data set represented complete dendritic morphologies (M), the third data set represented all physiological properties of OLM and HS cells (U). After the combined analyses of the morphological data (N), it was combined with the physiological data (U) to create complete cellular models (V). For further details see Methods. HS, hippocampo-septal; OLM, oriens-lacunosum-moleculare.

measured their properties at different points of the dendritic trees of OLM and HS cells. First, dendritic trees were reconstructed in 3D at the light microscopic level using the Neurolucida software. Then, we selected dendritic segments randomly so that dendritic segments at different distances from the somata are represented relatively evenly from the whole dendritic tree. Dendritic segments from every 100 micrometers from the somata were represented in the samples. We also ensured that every dendritic order was represented by several segments (Fig 2). The order of the selected segments and the distance of their starting point from the somata were determined using the dendrograms created by the Neurolucida Explorer software. Then, these sections were re-embedded and serially sectioned for electron microscopic reconstruction, and 53 OLM cell dendritic segments (median length: 15.7 µm) and 37 HS cell dendritic segments (median length: 11.1 µm) were fully reconstructed. Plasma membranes of the dendrites, outer membranes of mitochondria, and synaptic membranes were reconstructed in 3D and their dimensions were measured.

Proximal dendrites of OLM and HS cells were mostly smooth, whereas their distal dendrites carried sparsely distributed spine-like structures. On OLM cells, some of these

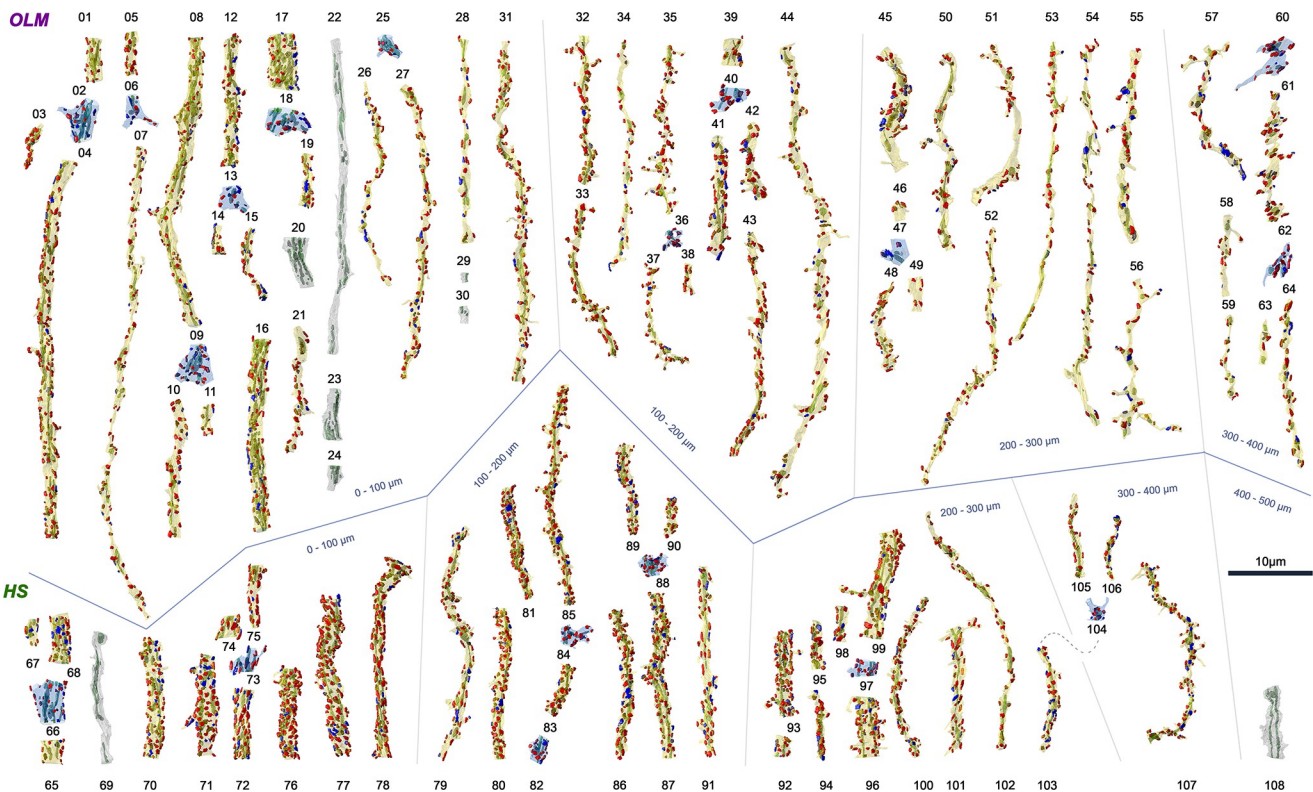

**Fig 2. 3D EM reconstructions of dendritic segments and dendritic branching points of OLM and HS cells.** 3D rendering of all reconstructed OLM dendritic (yellow or gray) and branching point (light blue) segments ($n = 53 + 11$, respectively), and all HS dendritic (yellow or gray) and branching point (light blue) segments ($n = 37 + 7$, respectively). Besides the dendrites (yellow) with fully reconstructed synapses ($n = 4,536$), there were some dendrites (gray) without synaptic reconstructions. Before ultracutting, OLM and HS cells were reconstructed under a light microscope using Neurolucida to determine the dendritic order and the distance of the sampled dendritic segments from the soma along the dendritic tree. The dendritic segments and branching points of OLM and HS cells are illustrated in groups in the upper and lower rows, respectively. Dendritic segments are arranged based on their distance from the soma. The dendritic membranes were made partially transparent to reveal their mitochondria (green) inside the dendrites and the input p-glutamatergic (red) excitatory ($n = 3,876$) and p-GABAergic (blue) inhibitory ($n = 660$) synapses on both sides of the dendrites. They were imaged in SEM and reconstructed from serial ultrathin sections immunostained for the GABAergic synapse marker gephyrin. Thickness of the dendrites varies greatly even within a given distance range. The surfaces of the HS cell dendrites are more densely covered by synapses than those of OLM cells. All raw morphological parameters are presented in S1–S5 Data files. EM, electron microscopic; HS, hippocampo-septal; OLM, oriens-lacunosum-moleculare.

protrusions were very long. The dendritic shafts received many inputs, most of which formed p-glutamatergic, type I (asymmetric) synapses (Figs 2 and 3). They were innervated by fewer p-GABAergic, type II (symmetric) inputs (Figs 2 and 3). Spines were different from PC spines as they did not have a typical head and were often innervated by more than one synaptic input. Here, sometimes, we refer to synapses established on OLM and HS cell dendrites as "inputs" to clearly differentiate them from synapses that the axons of these cells establish on other cells.

The following parameters were measured for each dendritic segment (S2–S4 Data): distance from the soma, length, dendritic order, volume with and without mitochondria, the volume of individual mitochondria, surface area, cross-sectional areas and perimeters of the dendrite along the dendritic segments, number of p-glutamatergic/p-GABAergic inputs, size of individual p-glutamatergic/p-GABAergic inputs, all surfaces of p-glutamatergic/p-GABAergic inputs including synapses on somata, dendritic shafts and spines. Morphological parameters normalized by surface area and length were also calculated.

### Identification of branching orders

Here, we defined dendritic orders in the following way: first-order dendrites originated from the somata, whereas the branching order of all offspring dendrites was one more than the order of the dendrite that they originated from (for example, all offspring of a second-order dendrite were third-order dendrites, and so on). However, although this ordering is unbiased, dendrites with the same order could be very different, partly because dendrites with the same order could have very different number of offspring dendrites that they supported. For example, we found that a first-order proximal dendrite could have the same thickness as a distal 10th-order dendrite. These data can be found in Table 1 and S2 and S3 Data files. Branching points (n = 11 OLM and 6 HS) of the dendritic trees were analyzed separately (Fig 2 and S4 and S5 Data files).

**Table 1. Comparisons of anatomical properties of OLM and HS cells (Mann–Whitney U test).**

| Variable | OLM/HS | n | Median | Lower quartile | Upper quartile | p = |
|---|---|---|---|---|---|---|
| Number of p-glutamatergic synapses/µm$^2$ dendritic surface | OLM | 47 | 0.522 | 0.408 | 0.599 | ***0.000000*** |
| | HS | 35 | 0.763 | 0.603 | 0.888 | |
| Number of p-GABAergic synapses/µm$^2$ dendritic surface | OLM | 47 | 0.075 | 0.042 | 0.114 | ***0.000001*** |
| | HS | 35 | 0.135 | 0.103 | 0.194 | |
| Total size of all p-glutamatergic synapses/µm$^2$ dendritic surface | OLM | 47 | 0.078 | 0.059 | 0.103 | ***0.000000*** |
| | HS | 35 | 0.121 | 0.095 | 0.158 | |
| Total size of all p-glutamatergic synapses/µm$^2$ dendritic surface | OLM | 47 | 0.015 | 0.006 | 0.021 | ***0.000008*** |
| | HS | 35 | 0.026 | 0.019 | 0.037 | |
| Proportion (%) of GABAergic inputs/all inputs (on segments with more than 10 inputs) | OLM | 42 | 14.3 | 9.0 | 20.5 | 0.303842 |
| | HS | 35 | 15.4 | 11.1 | 20.8 | |
| Size of individual p-glutamatergic synapses on proximal (0–100 µm) dendrites (µm$^2$) | OLM | 857 | 0.125 | 0.079 | 0.176 | ***0.000000*** |
| | HS | 604 | 0.164 | 0.114 | 0.232 | |
| Size of individual p-glut.ergic syn. on the middle part of the dendritic tree (100–250 µm) (µm$^2$) | OLM | 693 | 0.143 | 0.091 | 0.216 | ***0.028158*** |
| | HS | 1067 | 0.134 | 0.089 | 0.191 | |
| Size of individual p-glutamatergic synapses on distal (>250 µm) dendrites (µm$^2$) | OLM | 215 | 0.153 | 0.100 | 0.239 | ***0.000151*** |
| | HS | 202 | 0.129 | 0.086 | 0.176 | |
| Proportion of mitochondria (%) in dendritic segments and branching points | OLM | 64 | 9.3 | 7.2 | 11.3 | ***0.000228*** |
| | HS | 44 | 12.3 | 9.4 | 16.5 | |
| Soma volume (µm$^3$) | OLM | 3 | 1565 | 1541 | 1779 | 0.700000 |
| | HS | 3 | 1526 | 1450 | 2101 | |

*(Continued)*

**Table 1.** (Continued)

| Variable | OLM/HS | n | Median | Lower quartile | Upper quartile | p = |
|---|---|---|---|---|---|---|
| Nucleus volume ($\mu m^3$) | OLM | 3 | 536 | 486 | 640 | 0.700000 |
| | HS | 3 | 596 | 564 | 639 | |
| Soma surface area ($\mu m^2$) | OLM | 3 | 855 | 571 | 900 | 0.400000 |
| | HS | 3 | 972 | 768 | 973 | |
| Volume of mitochondria in soma ($\mu m^3$) | OLM | 3 | 83.2 | 65.8 | 83.9 | 0.700000 |
| | HS | 3 | 65.2 | 62.8 | 133.8 | |
| Volume of citoplasm. (= soma -nucl.-mito volume) of soma ($\mu m^3$) | OLM | 3 | 989 | 945 | 1056 | 0.700000 |
| | HS | 3 | 899 | 745 | 1371 | |
| Size of individual p-glutamatergic synapses on somata ($\mu m^2$) | OLM | 342 | 0.110 | 0.076 | 0.162 | ***0.000000*** |
| | HS | 527 | 0.072 | 0.051 | 0.105 | |
| Size of individual p-GABAergic synapses on somata ($\mu m^2$) | OLM | 52 | 0.242 | 0.131 | 0.488 | ***0.000000*** |
| | HS | 202 | 0.093 | 0.063 | 0.138 | |
| Size of individual axonal synapses ($\mu m^2$) | OLM | 134 | 0.102 | 0.071 | 0.153 | ***0.000000*** |
| | HS | 93 | 0.058 | 0.038 | 0.073 | |
| Volume of individual mitochondria in axons ($\mu m^3$) | OLM | 69 | 0.088 | 0.051 | 0.133 | ***0.000014*** |
| | HS | 54 | 0.046 | 0.030 | 0.060 | |
| Number of synapses/$\mu m$ axon | OLM | 20 | 0.404 | 0.208 | 0.494 | 0.373045 |
| | HS | 4 | 0.451 | 0.398 | 0.541 | |
| Number of mitochondria/$\mu m$ axon | OLM | 20 | 0.199 | 0.176 | 0.264 | 0.130926 |
| | HS | 4 | 0.265 | 0.231 | 0.318 | |
| Total dendritic length ($\mu m$) of completely reconstructed dendritic trees | OLM | 4 | 6367 | 4853 | 7182 | 0.193932 |
| | HS | 4 | 7484 | 7031 | 7813 | |
| Total number of nodes of completely reconstructed dendritic trees | OLM | 4 | 56.5 | 42.0 | 65.0 | 0.312322 |
| | HS | 4 | 43.0 | 38.5 | 46.0 | |
| Total number of dendritic endings of completely reconstructed dendritic trees | OLM | 4 | 61.0 | 46.5 | 68.5 | 0.383631 |
| | HS | 4 | 48.5 | 43.5 | 52.5 | |
| Number of first-order dendrites of completely reconstructed dendritic trees | OLM | 4 | 4.0 | 3.5 | 5.0 | 0.233780 |
| | HS | 4 | 5.5 | 4.5 | 6.0 | |

Underlined, italic *p-values* show significant differences.

HS, hippocampo-septal; OLM, oriens-lacunosum-moleculare.

## Dendritic thicknesses are highly variable

The general assumption about dendritic morphology is that the thickness of a dendrite decreases towards distal dendrites and towards higher order dendrites. We confirmed this view in both OLM and HS cells. However, we also found a very high variability of thickness (cross-sectional areas) within the same order of dendrites and at different distances both in our light microscopic (Neurolucida drawings) and electron microscopic (3D reconstructions) samples (Fig 2 and S2 and S3 Data files).

Morphologically detailed models are typically represented as interconnected cylinders or truncated cones, and these segments are characterized by their length and diameter (or the diameters at the 2 ends in the case of truncated cones). To aid the construction of such detailed model neurons, we created equivalent cylinder models for each of our electron microscopic samples of OLM and HS cell dendrites (with the exception of branch points). We calculated the diameter of the cylinder in 3 different ways: each cylinder had the same length as the 3D reconstructed segment, but one version had the same volume as the original segment, a second

version had the same external membrane surface, while the third one had the same average perimeter. These methods yielded different, but highly correlated estimates of the equivalent dendritic diameter (S17 and S18 Data files), and further confirmed our previous conclusions regarding the high variability of dendritic thickness.

### Thinner dendrites receive fewer synaptic inputs

After analyzing the number of synaptic inputs per dendritic length (linear density of synapses) on 47 OLM and 35 HS cell dendritic segments, we found that thinner dendrites (with smaller cross-sectional area), as expected, received fewer inputs per micrometer dendrite in both cell types. This was significant for both p-glutamatergic and p-GABAergic synaptic inputs of both cell types (Figs 2 and 3E and Tables 2 and 3, S2 and S3 Data files).

Because dendrites became thinner more distally from the soma, the linear density of p-glutamatergic dendritic synaptic inputs decreased significantly with the distance from the soma (Fig 3 and Table 3). The linear density of p-GABAergic synaptic inputs did not correlate with the distance from the soma in OLM cells but showed a weaker correlation with distance in HS cells (Fig 3G and Table 3).

The higher the branching order of the OLM dendrites were, the smaller linear density of p-glutamatergic inputs they received (Table 3), whereas the linear density of p-GABAergic inputs of HS cells showed a similar correlation with the branching order (Table 3).

Correlations between the linear densities and distance or branching order may partly be explained by the fact that the median thickness of the dendrites decreases away from the soma, towards higher order dendrites (although with very high variability).

### HS cell dendrites receive denser synaptic inputs than OLM cells

We compared the number of synaptic inputs of OLM and HS cells for unit surface area (surface density of synaptic inputs). We found that the surface density of synaptic inputs on HS dendrites was significantly higher than that for the OLM cells. OLM cell received 0.522

**Table 2. Comparison of anatomical properties within cell types (Mann–Whitney U test).**

| Variable | OLM/HS | n | Median | Lower quartile | Upper quartile | P |
|---|---|---|---|---|---|---|
| Proportion of mitochondria (%) in dendritic segments | OLM | 53 | 9.1 | 6.8 | 10.8 | **0.018827** |
| Proportion of mitochondria (%) in branching points | OLM | 11 | 9.6 | 9.3 | 56.8 | |
| Proportion of mitochondria (%) in dendritic segments | HS | 37 | 12.1 | 9.2 | 14.7 | **0.000077** |
| Proportion of mitochondria (%) in branching points | HS | 7 | 29.6 | 19.9 | 43.9 | |
| Size of individual p-glutamatergic synapses on somata ($\mu m^2$) | OLM | 342 | 0.110 | 0.076 | 0.162 | **0.000000** |
| Size of individual p-GABAergic synapses on somata ($\mu m^2$) | OLM | 52 | 0.242 | 0.131 | 0.488 | |
| Size of individual p-glutamatergic synapses on somata ($\mu m^2$) | HS | 527 | 0.072 | 0.051 | 0.105 | **0.000000** |
| Size of individual p-GABAergic synapses on somata ($\mu m^2$) | HS | 202 | 0.093 | 0.063 | 0.138 | |
| Size of individual p-glutamatergic synapses on somata ($\mu m^2$) | OLM | 342 | 0.110 | 0.076 | 0.162 | **0.029445** |
| Size of individual p-glutamatergic synapses on proximal (0–50 $\mu$m) dendrites ($\mu m^2$) | OLM | 857 | 0.125 | 0.079 | 0.176 | |
| Size of individual p-glutamatergic synapses on somata ($\mu m^2$) | HS | 527 | 0.072 | 0.051 | 0.105 | **0.000000** |
| Size of individual p-glutamatergic synapses on proximal (0–50 $\mu$m) dendrites ($\mu m^2$) | HS | 181 | 0.150 | 0.104 | 0.208 | |
| Size of individual p-GABAergic synapses on somata ($\mu m^2$) | OLM | 52 | 0.242 | 0.131 | 0.488 | **0.000302** |
| Size of individual p-GABAergic synapses on proximal (0–50 $\mu$m) dendrites ($\mu m^2$) | OLM | 124 | 0.134 | 0.085 | 0.237 | |
| Size of individual p-GABAergic synapses on somata ($\mu m^2$) | HS | 202 | 0.093 | 0.063 | 0.138 | **0.000000** |
| Size of individual p-GABAergic synapses on proximal (0–50 $\mu$m) dendrites ($\mu m^2$) | HS | 116 | 0.166 | 0.103 | 0.234 | |

Underlined *p-values* show significant differences.

HS, hippocampo-septal; OLM, oriens-lacunosum-moleculare.

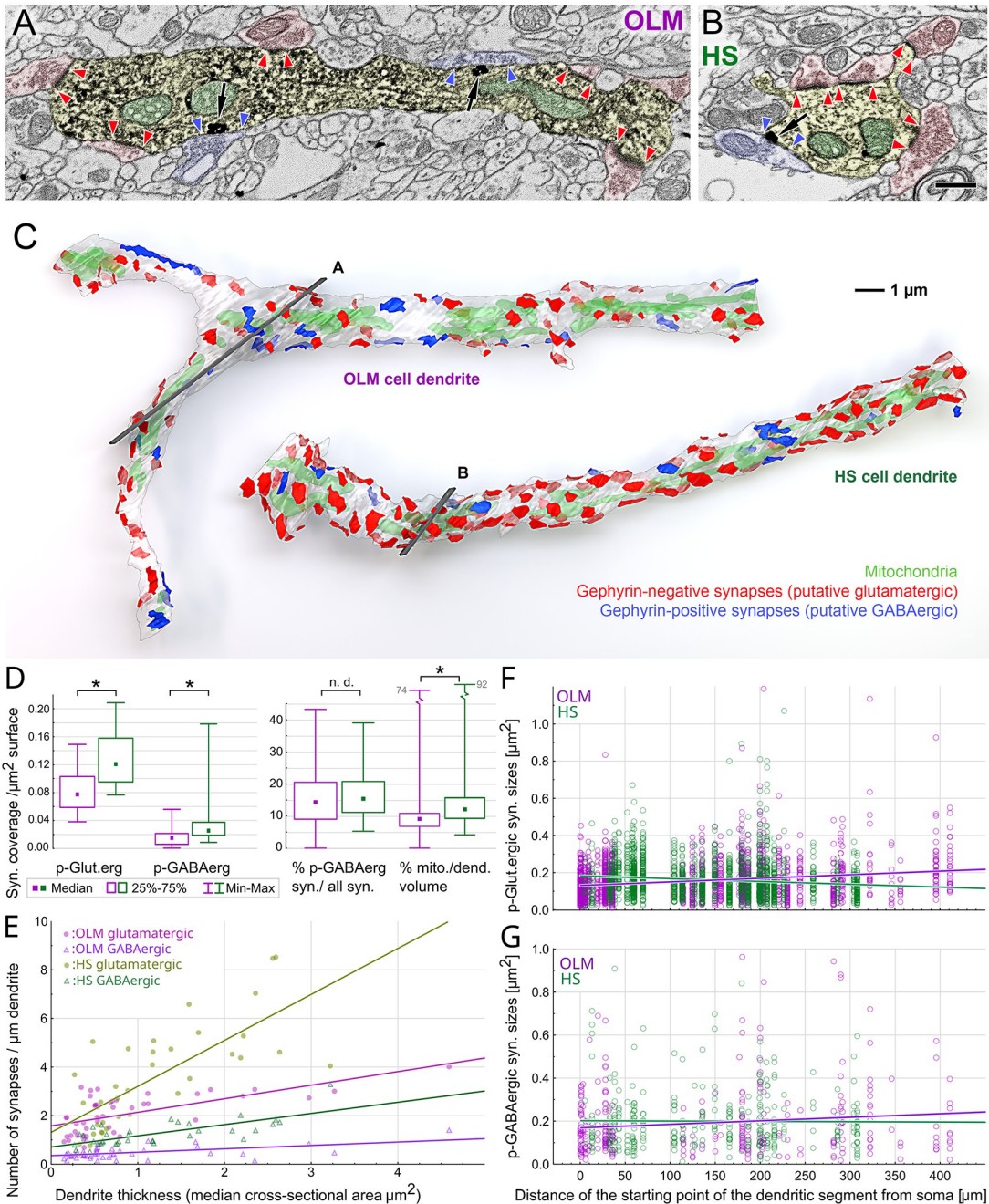

**Fig 3. HS cell dendrites receive more synaptic inputs and have larger mitochondrial volume than OLM cell dendrites.** (A–C) Comparison of a typical OLM- and a typical HS dendritic segment of similar diameters. (A, B) Artificially colored scanning electron microscopic images of the DAB–labeled dendrites (yellow) of an OLM (A) and an HS cell (B). Preembedding immunogold labeling of gephyrin (black particles, black arrows) was used to distinguish between GABAergic and glutamatergic synapses. Red and blue arrowheads label the edges of gephyrin-negative and gephyrin-positive synapses of p-glutamatergic (red) and p-GABAergic (blue) boutons, respectively. Mitochondria are colored green in the electron micrographs. (C) 3D reconstructions of the dendritic segments shown in A and B. The section plane of A and B are indicated by gray in C. The OLM and HS dendritic segments in C are also shown in Fig 2 (dendrite 12–15 and dendrite 78, respectively). Putative glutamatergic and p-GABAergic synapses are shown in red and blue, respectively. The dendritic membranes (white) were made partially transparent to reveal mitochondria (green) inside the dendrites and synapses on the other side of the dendrites. Both cell types have irregularly shaped dendritic shafts that are sparsely spiny. (D) Ultrastructural parameters of reconstructed OLM and HS dendritic segments show characteristic differences. Medians (square) and interquartile ranges (boxes), minimum and maximum values (whiskers) of measured parameters are shown. Asterisks indicate significant differences, n.d: no significant difference was found. See medians, interquartile ranges, and *p*-values in Table 1 and raw data in S2–S5 Data files. The portions of surfaces

covered by p-glutamatergic and p-GABAergic synapses are significantly larger on HS cell dendrites than on OLM cell dendrites. There is no difference between the ratios of GABAergic inputs on the dendrites of the 2 populations of interneurons. Mitochondria of HS cell dendrites occupy significantly larger volume inside the dendrites than those of OLM cells. (E) The linear density of p-glutamatergic (filled circles) and p-GABAergic (open triangles) synapses show positive correlation (Spearman rank) with the thickness of the dendrite both in the case of OLM (purple) and HS cells (green). However, the linear densities are markedly different. (F) The sizes of individual p-glutamatergic synapses varied to a large extent, and they showed a significant correlation with the distance from the soma along the dendritic tree in both cell groups. Input synapses of OLM cells became larger, whereas inputs of HS cells became smaller with the distance from the somata. (G) The p-GABAergic inputs showed similar trends without being statistically significant. For E–G, p-values can be found in Table 3. Raw morphological parameters are presented in S2–S5 Data files. HS, hippocampo-septal; OLM, oriens-lacunosum-moleculare.

p-glutamatergic and 0.075 p-GABAergic synapses/$\mu m^2$, whereas HS cells were innervated by 0.763 p-glutamatergic and 0.135 p-GABAergic synapses/$\mu m^2$ (medians; for interquartile ranges, see Table 1). Consequently, synaptic inputs proportionally covered significantly larger areas on HS dendrites (Fig 3D and Table 1). These differences were detected for both p-glutamatergic and p-GABAergic synaptic inputs (Table 1).

## Distribution of glutamatergic and GABAergic synapses on dendritic shafts and spines

OLM and HS cells had sparsely distributed spines on their dendrites. On the proximal dendrites (dendrites closer than 100 μm from the soma), only 6% and 4.5% of the excitatory and 2.7% and 0% of the inhibitory inputs targeted the spines of OLM and HS cells, respectively. More distally ($>$100 μm), spine synapses became more frequent. On distal dendrites, 23.9% and 12.9% of excitatory synaptic inputs and 16.6% and 7.8% of inhibitory synaptic inputs innervated spines of OLM and HS cells, respectively.

On OLM cell dendrites, spine synapses (p-Glut: 0.123 $\mu m^2$, p-GABA: 0.090 $\mu m^2$) were significantly smaller (p-Glut: $p = 0.0011$, p-GABA: $p = 0.0398$) than dendritic shaft synapses (p-Glut: 0.141 $\mu m^2$, p-GABA: 0.137 $\mu m^2$). On HS cell dendrites, p-glutamatergic spine synapses (0.109 $\mu m^2$) were significantly smaller ($p < 0.0001$) than shaft synapses (0.146 $\mu m^2$). However, p-GABAergic spine synapses (0.147 $\mu m^2$) were not significantly different from shaft synapses (0.163 $\mu m^2$) of p-GABAergic terminals on HS cells (S19 Data).

## HS cell dendrites had relatively more mitochondria than OLM cells

Thicker dendrites (with larger median cross-sectional area) were occupied by proportionally larger mitochondria for both OLM and HS cells. The correlation was significant (Table 3). This mitochondrial ratio was similar along the proximal to distal dendrites; therefore, we found no correlation between mitochondrial volume density (%) and the somatic distance or the order of the dendritic segments (Table 3).

However, HS cells had a significantly larger proportion of mitochondrial volumes compared to OLM cells (median 12.3% versus 9.3% in HS versus OLM cells, respectively; Fig 3D and Table 1).

Mitochondria occupied a significantly larger proportion of the cytosol in the branching points compared to the dendritic shafts (Table 2). This is likely due to the fact that branching points are naturally thicker in the dendrites; therefore, they more likely had enough space for mitochondria than thinner dendrites.

## Proportions of p-GABAergic synaptic inputs do not change along the dendrites and do not differ between OLM and HS cells

We found no correlation between the ratio of p-GABAergic inputs (out of the sum of all glutamatergic and GABAergic inputs) and either their distance from the soma or the dendritic

**Table 3. Correlation between anatomical properties within cell types (Spearman rank correlations).**

| Correlation between these 2 variables | | Cell type | n | Spearman | p-Value |
|---|---|---|---|---|---|
| Dendritic order | Thickness of segment (average diameter in μm; Neurolucida drawing) | OLM | 444 | *−0.578218* | *0.000000* |
| | | HS | 361 | *−0.604258* | *0.000000* |
| Dendritic order | Median of dendrite cross-sectional area (μm$^2$) | OLM | 53 | *−0.714852* | *0.000000* |
| | | HS | 37 | *−0.633558* | *0.000026* |
| Distance from soma (μm) | Median of dendrite cross-sectional area (μm$^2$) | OLM | 53 | *−0.531898* | *0.000042* |
| | | HS | 37 | *−0.452745* | *0.004898* |
| Median of dendrite cross-sectional area (μm$^2$) | Number of p-glutamatergic synapses/μm dendrite | OLM | 47 | *0.639801* | *0.000001* |
| | | HS | 35 | *0.735014* | *0.000000* |
| Median of dendrite cross-sectional area (μm$^2$) | Number of p-GABAergic synapses/μm dendrite | OLM | 47 | *0.354507* | *0.014489* |
| | | HS | 35 | *0.675070* | *0.000009* |
| Distance from soma (μm) | Number of p-glutamatergic synapses/μm dendrite | OLM | 47 | *−0.610393* | *0.000005* |
| | | HS | 35 | *−0.422306* | *0.011500* |
| Distance from soma (μm) | Number of p-GABA synapses/μm dendrite | OLM | 47 | −0.030555 | 0.838448 |
| | | HS | 35 | *−0.363598* | *0.031789* |
| Dendritic order | Number of p-glutamatergic synapses/μm dendrite | OLM | 47 | *−0.527713* | *0.000138* |
| | | HS | 35 | −0.306232 | 0.073589 |
| Dendritic order | Number of p-GABAergic synapses/μm dendrite | OLM | 47 | −0.222336 | 0.133075 |
| | | HS | 35 | *−0.408740* | *0.014774* |
| Distance from soma (μm) | % GABAergic inputs/all inputs | OLM | 47 | 0.280228 | 0.056412 |
| | | HS | 35 | −0.089540 | 0.608990 |
| Dendritic order | % GABAergic inputs/all inputs | OLM | 47 | 0.067832 | 0.650525 |
| | | HS | 35 | −0.191839 | 0.269589 |
| Distance of soma (μm; den starting point) | Individual p-glutamatergic syn size (μm$^2$) | OLM | 1765 | *0.178743* | *0.000000* |
| | | HS | 1874 | *−0.154737* | *0.000000* |
| Dendritic order | Individual p-glutamatergic syn size (μm$^2$) | OLM | 1765 | *0.101080* | *0.000021* |
| | | HS | 1874 | *−0.144467* | *0.000000* |
| Distance of soma (μm, den starting point) | Individual p-GABAergic syn size (μm$^2$) | OLM | 287 | 0.049482 | 0.403640 |
| | | HS | 324 | 0.021623 | 0.698194 |
| Dendritic order | Individual p-GABAergic syn size (μm$^2$) | OLM | 287 | −0.086126 | 0.145558 |
| | | HS | 324 | 0.046594 | 0.403213 |
| Mitochondria volume/μm dendrite | Median of dendrite cross-sectional area (μm$^2$) | OLM | 53 | *0.820916* | *0.000000* |
| | | HS | 37 | *0.775249* | *0.000000* |
| Mitochondria % in dendrite (volume) | Distance from soma (μm) | OLM | 53 | −0.233406 | 0.092564 |
| | | HS | 37 | *0.349816* | *0.033805* |
| Mitochondria % in dendrite (volume) | Dendritic order | OLM | 53 | −0.231975 | 0.094639 |
| | | HS | 37 | 0.266902 | 0.110288 |

Underlined *values* show significant differences.

HS, hippocampo-septal; OLM, oriens-lacunosum-moleculare.

orders of their OLM or HS cells (Table 3). The median ratio of p-GABAergic inputs (on dendritic segments with at least 10 synaptic inputs) was not different (14.3% for OLM and 15.4% for HS cells; Fig 3D and Table 1). Nevertheless, because HS cells receive denser synaptic inputs they received a significantly larger number of p-GABAergic and p-glutamatergic inputs (see above).

## More distal OLM dendrites receive larger p-glutamatergic inputs while this correlation is the opposite for HS cells

We reconstructed 1,765 and 1,874 p-glutamatergic inputs on OLM and HS dendrites, respectively, and analyzed the correlation between their size and their distance from the soma (somatic distance). The somatic distance of all synaptic inputs on the same dendritic segment was considered identical and equal to the somatic distance of the dendritic segment. We used this simplification because the median dendritic segment length was very small (OLM: 15.7, HS: 11.1 μm) compared to the up to 432 μm long dendrites examined. The somatic distance of the dendritic segment was measured along the dendrite from the soma to the starting point of the given dendritic segment.

Although the size of individual synapses varied to a very large extent, we found a strong correlation between the somatic distances and the sizes of the synapses. The sizes of p-glutamatergic inputs increased on average towards more distal dendrites on OLM cells, whereas synaptic size decreased with increasing distance from the soma on HS cells (Fig 3F and Table 3). Putative glutamatergic inputs of proximal (0 to 100 μm) OLM dendrites were significantly smaller (median: 0.125 versus 0.164 μm$^2$), whereas distal (farther than 250 μm) dendrites were significantly larger (median: 0.153 versus 0.129 μm$^2$) than those on HS cells (Table 1). Similar significant correlations were found between the size of p-glutamatergic synapses and the order of the dendritic segments (Table 3).

Although data suggested a similar trend for GABAergic inputs (Fig 3G), the same analysis revealed significant correlations neither for the distance, nor for the order of the dendritic segments (Table 3). This may have been due to the smaller numbers of p-GABAergic synapses (284 and 324 inputs on OLM and HS cells, respectively).

## Inputs of OLM and HS cell somata

Somata of OLM and HS cells were reconstructed and analyzed similarly to the dendrites. The total somatic surfaces and volumes of OLM and HS cells were similar ($n$ = 3 OLM and 3 HS cells; Fig 4 and Table 1 and S6 Data). Somatic inputs were reconstructed from other partially (about 25% to 60%) reconstructed somata ($n$ = 4 OLM and 4 HS cell bodies, Fig 4). OLM and HS somata received many p-glutamatergic inputs and about 8% to 39.7% p-GABAergic inputs (S6 Data). Using the total somatic surface and the surface density of input synapses, we estimated that an OLM cell soma receives about 173 glutamatergic and 21 GABAergic inputs, whereas an HS cell soma receives 414 glutamatergic and 147 GABAergic inputs.

We found p-GABAergic inputs to be significantly larger than p-glutamatergic inputs on both cell types (Table 2). Both types of inputs were significantly larger on OLM cells compared to those on HS cells (Table 1).

We then compared somatic and dendritic synapses. Because dendritic inputs of OLM an HS cells correlated with somatic distance, we compared the synapses of only the proximal dendrites (0 to 100 μm) with the somatic ones.

Glutamatergic synapses of HS cells were significantly larger on proximal dendrites than on somata (Table 2). This difference was much smaller and less significant in the case of OLM cells (Table 2). GABAergic synapses were also larger on the dendrites of HS cells than their somatic synapses (Table 1). We found an opposite difference in OLM cells: GABAergic synapses were larger on the somata than on the dendrites (Table 2).

## Local postsynaptic targets of OLM cells are predominantly PC dendrites

The target-selectivity of OLM and HS cells has been investigated in rats [29,36,44]. Here, we investigated it in mice. Due to the high number of strongly labeled OLM cells, a very dense

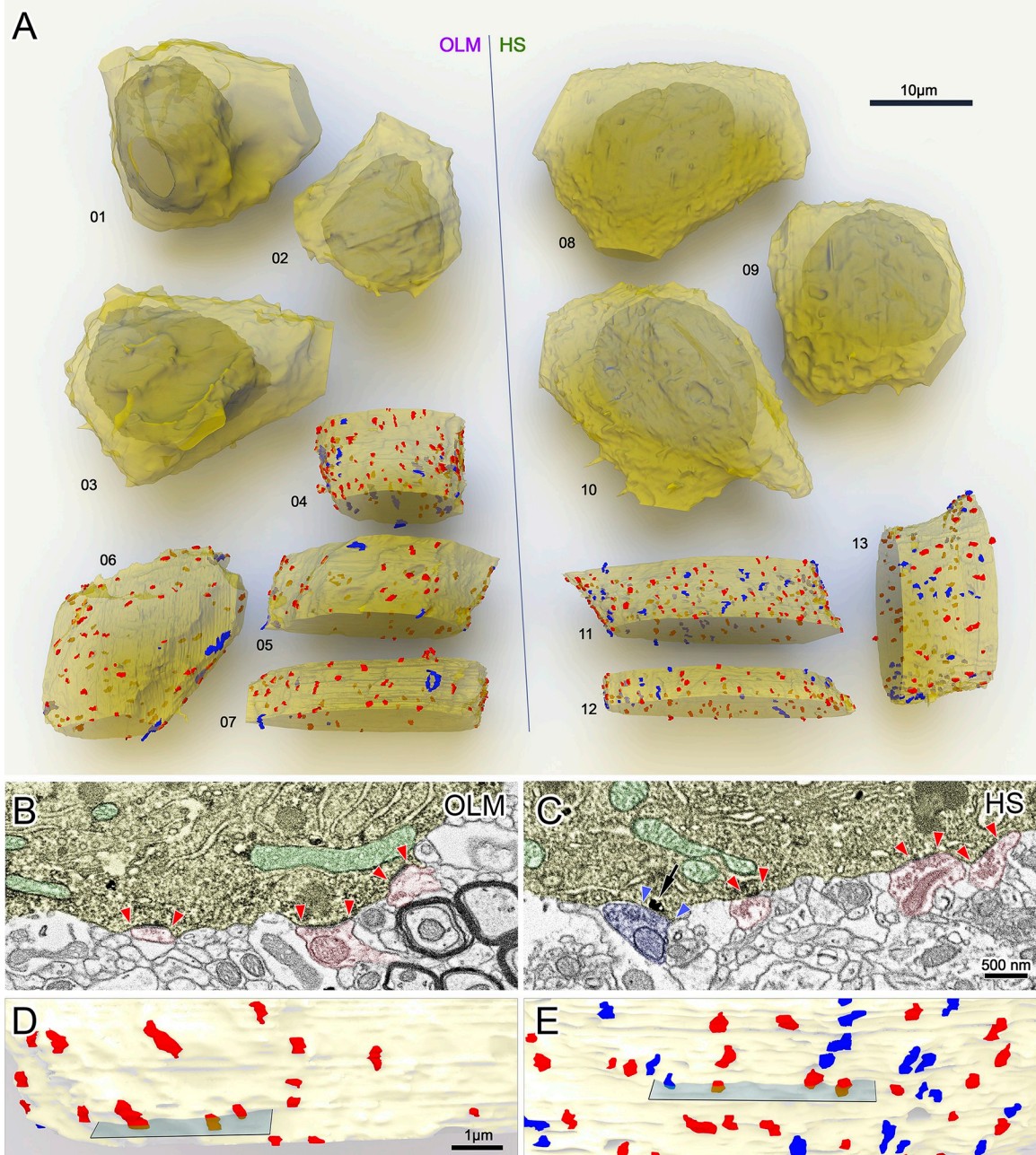

**Fig 4. 3D EM reconstructions of OLM and HS cell bodies.** (A) Whole somata of tracer-labeled cells were reconstructed to estimate the total somatic surfaces and volumes of OLM and HS cells (OLM: 01–03, HS: 08–10). Partial segments of somata (OLM: 06–07, HS: 11–13) were sampled to estimate the parameters of p-glutamatergic (red) and p-GABAergic (blue) synaptic inputs. The dendritic membranes (yellow) were made partially transparent to reveal the nuclei (whole somata) and synapses on the other side of the somata. (B–E) Representative sections of the reconstructed somatic membranes of an OLM (B, D) and an HS (C, E) cell. (B, C) Artificially colored scanning electron micrographs of the DAB–labeled cell bodies (yellow) of an OLM (B) and an HS cell (C). Preembedding immunogold labeling of gephyrin (black particles, arrow) was used to distinguish between GABAergic and glutamatergic synapses. Red and blue arrowheads label the edges of p-glutamatergic (red) and p-GABAergic (blue) synapses, respectively. Mitochondria are colored green in the electron micrographs. (D, E) Parts of the 3D reconstructions of the cells and their input synapses. Sections from B and C are indicated by gray planes. Also see Tables 1 and 2 and S6 Data. EM, electron microscopic; HS, hippocampo-septal; OLM, oriens-lacunosum-moleculare.

axonal labeling was found in str. lacunosum-moleculare (Fig 5A). Using scanning electron microscopic array tomography, serially sectioned, randomly selected, tracer-labeled axonal segments were 3-dimensionally reconstructed (Fig 5B).

Postsynaptic targets of axons were followed in the serial images and were identified as described previously [61]. Briefly, spiny dendritic shafts without type 1 synaptic inputs on the shaft were considered PC dendritic shafts, whereas dendritic shafts that received type I inputs were identified as interneurons. Dendritic spines were identified by their size and specific shape emerging from a dendrite. Out of 136 OLM (from 2 mice) synapses, 78% and 19% targeted PC shafts and spines, respectively. These PC spines always received an unlabeled asymmetric input as well, and 1% of the synapses targeted interneuronal dendritic shafts; 2% of the targets could not be classified.

## Local postsynaptic targets of HS cells are predominantly PC dendrites

BDA injection into the MS revealed only a few HS cells with extensive labeling of their local axon collaterals in the mouse CA1 area. However, anterograde filling of the hippocampal axon collaterals of septo-hippocampal neurons was also unavoidable. Therefore, to make sure that we investigate the target selectivity of only HS axonal segments in CA1, we partially reconstructed the local axonal arbors of BDA-labeled HS cells using Neurolucida (MicroBright-Field). We analyzed these segments using correlative light and electron microscopy (Figs 5E–5K and S1).

The 3 reconstructed BDA-labeled HS cells had local axonal branches that arborized mostly in str. oriens and radiatum and did not enter str. lacunosum-moleculare (Figs 5E and S1). Out of 95 HS ($n$ = 3 cells from 2 mice) synapses 73% and 23% targeted PC shafts and spines, respectively. These PC spines always received an unlabeled asymmetric input as well, and 2% of the synapses targeted interneuronal dendritic shafts; 2% of the targets could not be classified.

## Morphologies of OLM and HS cell axons

We also investigated the morphologies of OLM cell axons to further facilitate their functional modeling (Fig 5B–5K and S7 Data). Axon terminals (en passant boutons) of these neurons were barely distinguishable in thickness from the intervaricose axon segments. We found that OLM cell axons establish 4 synapses per 10 μm axons, whereas HS cell axons establish 4.5 synapses per 10 μm axons. The sizes of output synapses of OLM cells were significantly larger (almost 2-fold difference) than those of HS cell axons (Table 1). OLM and HS cells had 2 and 2.7 pieces of mitochondria per 10 μm axon, respectively. The sizes of individual mitochondria in OLM axons were also significantly larger (2-fold difference) than those of HS cells (Table 1).

## Proportions of OLM and HS cells within the somatostatin-positive interneurons of CA1 str. oriens

We estimated the proportions of OLM and HS cells within somatostatin-positive interneurons in CA1 str. oriens using immunochemistry. In mice with the most effective tracer-labeling, minimum 49.1%, 42%, and 37.9% of somatostatin-positive cells were OLM cells (out of 1036 SOM-positive cell from 3 mice). Since the expression of the Chrna2 gene is not detectable in every anatomically identified OLM cell [62], and the AAV virus we used for identification in this study does not necessarily infect all Cre-positive cells at the injection site, this number should strictly be considered as an absolute minimum value. We found that minimum 9.5%, 8.1%, and 7.5% of SOM-positive cells were HS cells (out of 1,276 SOM-positive cells from 3 mice). The latter should also be considered as an absolute minimum proportion, as it is not possible to label all the cells projecting to the MS with retrograde tracers.

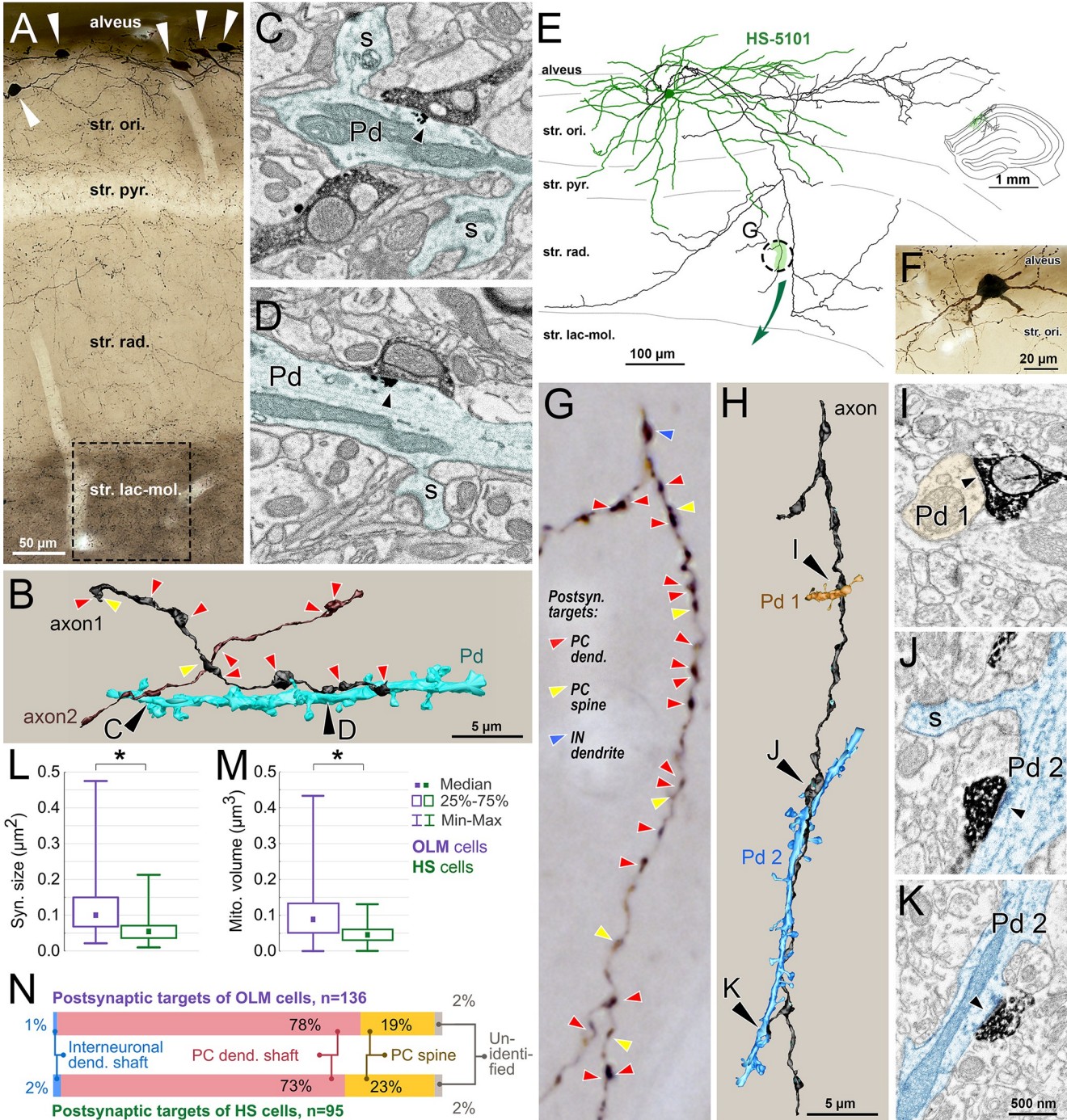

**Fig 5. Postsynaptic target analysis of OLM and HS cells.** For the explanation of the workflow, see Fig 1I and 1J. (A) Light microscopic image of the sampled area used for OLM cell axon reconstruction (dashed line) in the str. lacunosum-moleculare of the CA1 area in a Cre-dependent viral tracer-injected Chrna2-Cre mouse. Cell bodies of labeled OLM cells in the str. oriens are indicated by white arrowheads. (B) 3D scanning electron microscopic reconstruction of 2 OLM axonal segments (brown and black) and one of their common target pyramidal dendrites (turquoise). Postsynaptic targets of other synapses of the reconstructed axons are indicated by arrowheads (red: PC dendritic shaft, yellow: spine). (C, D) Electron micrographs of the synapses (black arrowheads) with the reconstructed pyramidal dendrite (Pd, turquoise). S labels spines. Immunogold labeling for gephyrin (black particles) is visible at the postsynaptic sites of the synapses. (E) Neurolucida reconstruction and (F) light microscopic image of the same BDA-labeled HS cell. The complete dendritic tree (green) and partial axonal arbor (black) of this cell were reconstructed. Inset shows the location of the cell in the hippocampus. (G) Light microscopic image of the axonal segment indicated with dashed circle in E. Postsynaptic targets of synapses are indicated on the correlated light microscopic image in G, as above. (H) The same axonal segment (black) was serially sectioned and reconstructed in 3D using an array scanning electron microscope. Two of the postsynaptic PC dendrites (Pd 1, orange and Pd 2, blue) were also reconstructed and shown in H. (I–K) Electron micrographs of synapses (arrowheads) of

the labeled HS cell axon with the reconstructed PC dendrites (orange and blue). The "blue" dendrite receives 2 synapses (J and K) from different branches of the reconstructed axon. Sections for HS cell axon reconstruction were not immunostained against gephyrin (see Fig 1 and Methods). (L) Individual synapses of OLM axons are significantly larger than those of HS cells. (M) Individual mitochondria are significantly larger in OLM cell axons than in HS cell axons. (N) Proportions of different postsynaptic targets in CA1 of OLM and HS cells. Medians, interquartile ranges, and *p*-values for L and M can be found in Table 1. Raw data are presented in S7 Data. Scale bar in K applies to all electron microscopic images. HS, hippocampo-septal; OLM, oriens-lacunosum-moleculare; PC, pyramidal cell.

## Morphological parameters of fully reconstructed dendritic trees of OLM and HS cells

Dendritic trees of 4 OLM and 4 HS cells were completely reconstructed from series of light microscopic sections of optimally fixed brains in 3D using Neurolucida (Fig 6). Reconstructions were adjusted for tissue shrinkage/dilatation that might have occurred during tissue processing (see correction factors in Methods). Both cell types had a large dendritic tree restricted to str. oriens/alveus. Their dendrites extended over 10 to 17, 60 μm thick sections. The total dendritic length, the total number of nodes and endings, and the number of first-order dendrites were not different between the 2 populations (Table 1). Sholl analysis was carried out to test whether there is a difference between OLM and HS cells in the number of dendrites intersecting spheres, the length, surface area, volume and average diameter of dendritic segments, the number of dendritic branching points, and endings in spheric shells (S4 Fig). This analysis did not reveal any significant differences, except that HS cells had branches in shells further than 500 μm, while the dendrites of the OLM cells did not reach that far from their somata.

## Electrophysiological properties of OLM and HS cells

We analyzed the membrane potential responses of 8 OLM and 8 HS cells (using a total of 46 measurements) to a series of current step injections during somatic whole-cell recordings in acute hippocampal slices (Fig 7, see Methods). From each recording, we extracted a total of 267 electrophysiological features corresponding to several standardized current amplitudes. We found that several of these features had significantly different values in the 2 cell types (Mann–Whitney test with Bonferroni correction to account for multiple comparisons, $p < 0.00018$; Table 4 and S11 Data). HS cells fired more rapidly at the beginning of large positive current injections, while OLM cells fired action potentials of larger amplitudes. However, we also found many features (including input resistance, the size of the hyperpolarizing sag response, the mean firing frequency, and action potential width) that were not significantly different between the 2 cell populations. More specifically, it is interesting to note that, although the spiking responses of the 2 cell types were clearly distinguishable, their subthreshold responses were quite similar.

## Modeling synaptic distributions in OLM and HS neurons

The anatomical and electrophysiological data collected in our study provide essential information for the construction of detailed models of hippocampal neurons and networks (see Discussion). As an initial demonstration of the utility of these data sets, we built and simulated a set of morphologically detailed models with a limited set of biophysical mechanisms to capture the subthreshold behavior of OLM and HS cells, focusing on the effects of excitatory and inhibitory synaptic input.

As a first step, we computed the number and spatial distribution of excitatory and inhibitory synapses on OLM and HS cells by combining our light microscopy-based reconstructions with our EM data. To this end, we examined the relationship between the density of synapses and several potential predictors such as dendritic diameter, distance from the soma, and dendritic order using multiple linear regression. We found that, overall, the density (number per

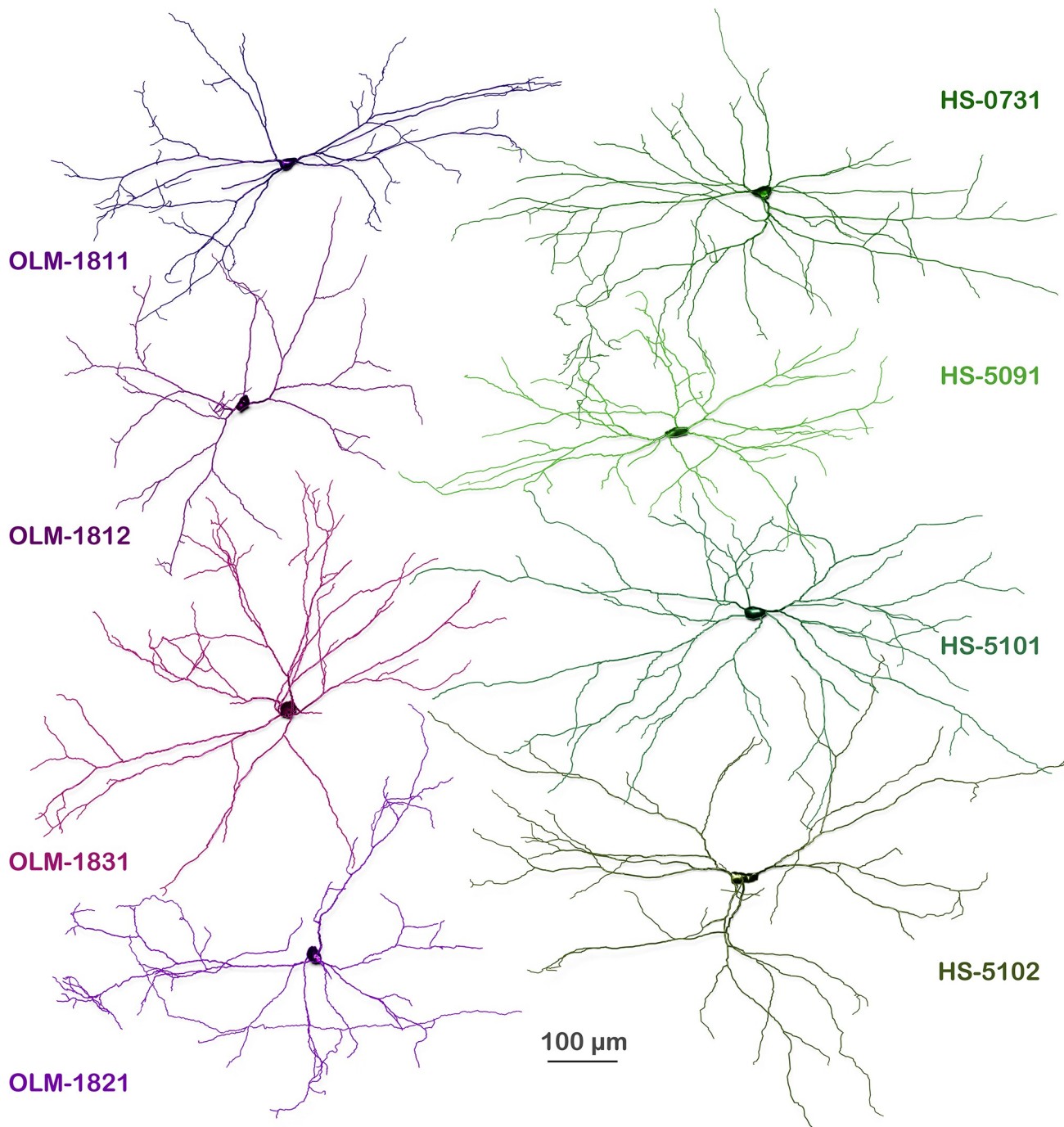

**Fig 6. Complete reconstruction of dendritic trees of OLM and HS cells.** Neurolucida reconstructions of cells from 10–17, 60 μm thick sections. All dendrites were in str. oriens and alveus. OLM cells are shown in different shades of purple. HS cells are shown in shades of green. The dendrites mostly arborized in a plane that was parallel with the border of the str. oriens and alveus. Here, all neurons are illustrated from a view perpendicular to that plane. These full reconstructions were the foundations (Fig 1G, 1L and 1M) of the computational models created in the modelling part of this study (Fig 1N and 1V). Dendrite segment analysis of the cells can be found in S15 Data. Sholl analysis of the cells are shown in S3 Fig and S20 Data. Three-dimensional Neurolucida reconstructions and dendrograms of the cells are presented in S21 and S22 Data files, respectively. HS, hippocampo-septal; OLM, oriens-lacunosum-moleculare.

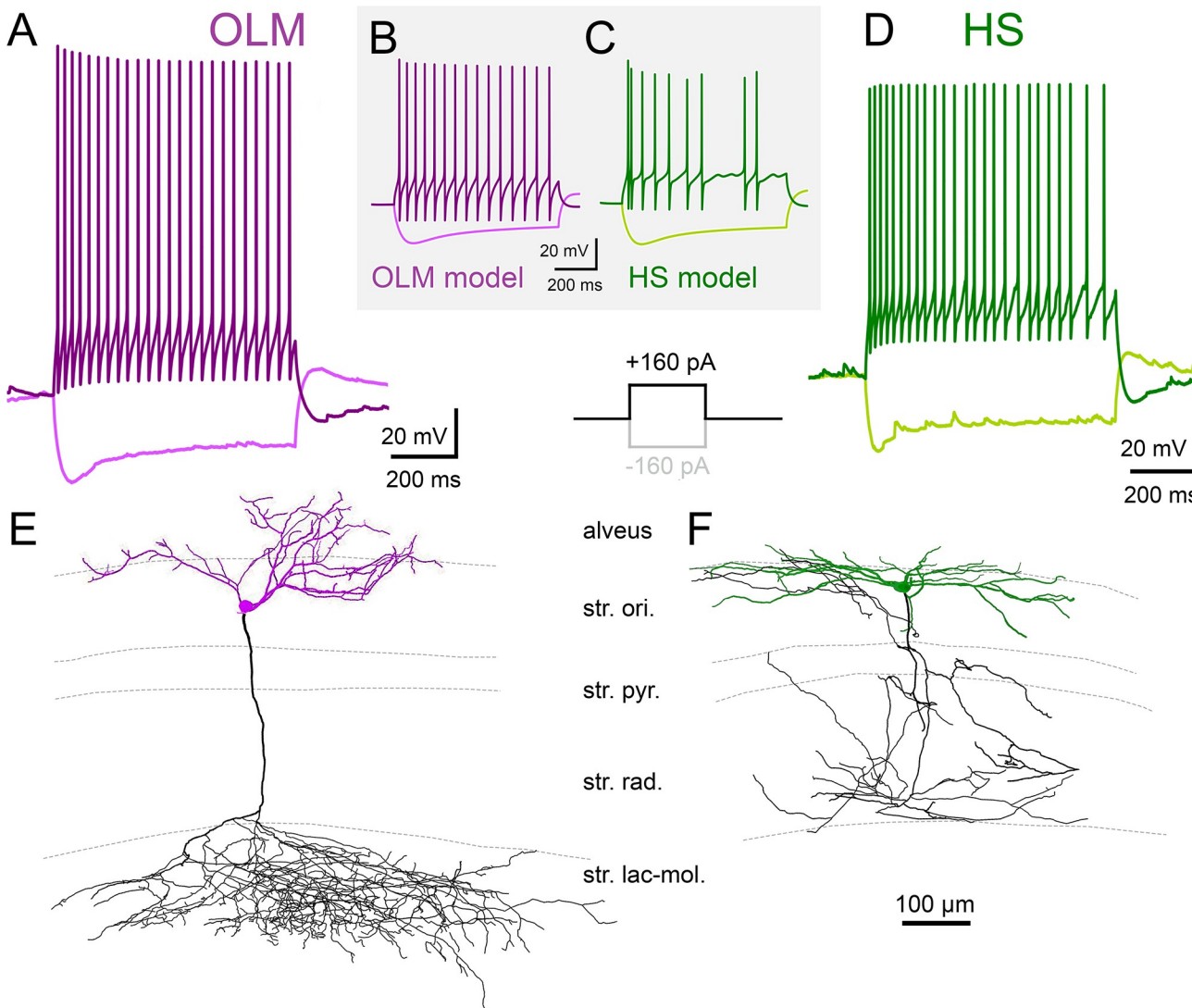

**Fig 7. Electrophysiological properties of OLM and HS cells.** For the explanation of the in vitro experimental workflow, see Fig 1O and 1U. (A, D) Typical voltage responses of OLM and HS cells to depolarizing and hyperpolarizing current steps (amplitude: 160 pA). (B, C) Example voltage responses from an active OLM and an active HS cell model, corresponding to the same stimulation protocol used in the experiments. (E, F) Reconstructions of a biocytin labeled OLM (E) and an HS cell (F). Dendrites are shown in purple and green, whereas axon arbors are shown in black. Cells were pre-labeled for in vitro recordings by Cre-dependent tracer virus injected in the CA1 area of Chrna2-Cre mice (OLM) or Fluospheres injected into the medial septum (HS cells). HS, hippocampo-septal; OLM, oriens-lacunosum-moleculare.

unit membrane area) of excitatory and inhibitory synapses in both cell types depended only weakly on each of these morphological variables, but the most reliable predictions could be made by considering the diameter of the dendrite and its distance from the soma (see Methods, Eq 1). There were also substantial differences in the prediction coefficients based on cell type (OLM versus HS) and synapse type (excitatory versus inhibitory) (S12 Data), consistent with the differences in average synapse density and in correlations with morphological features, as we described above (Table 3).

The predictive equations from multiple linear regression allowed us to estimate the density and, consequently, the number of excitatory and inhibitory synapses, in each segment of the fully reconstructed morphologies (4 OLM and 4 HS cells). Analyzing the resulting

**Table 4. Physiological features extracted from recordings and their statistics.**

| Feature | OLM/HS | Median | Lower quartile | Upper quartile | P (Mann–Whitney U test) |
|---|---|---|---|---|---|
| First inverse interspike interval (maxpsike) | OLM | 107.54 | 96.15 | 117.65 | **0.000000** |
| | HS | 169.49 | 153.85 | 196.38 | |
| Slow after hyperpolarization time (steady state) | OLM | 0.07 | 0.06 | 0.08 | **0.000000** |
| | HS | 0.13 | 0.09 | 0.15 | |
| Second inverse interspike interval (maxspike) | OLM | 104.17 | 99.75 | 113.64 | **0.000000** |
| | HS | 153.85 | 146.06 | 178.63 | |
| Peak of second AP (maxspike) | OLM | 31.15 | 28.68 | 37.67 | **0.000000** |
| | HS | 17.13 | 13.78 | 24.19 | |
| Amplitude of first AP (maxspike) | OLM | 96.40 | 94.40 | 102.47 | **0.000000** |
| | HS | 84.60 | 80.77 | 88.85 | |
| Third inverse interspike interval (maxspike) | OLM | 103.63 | 96.85 | 106.67 | **0.000001** |
| | HS | 147.06 | 129.87 | 170.95 | |
| Coefficient of variation of the interspike intervals (maxspike) | OLM | 0.12 | 0.08 | 0.17 | **0.000001** |
| | HS | 0.24 | 0.18 | 0.29 | |
| Average height of AP's (maxspike) | OLM | 26.60 | 24.79 | 28.89 | **0.000001** |
| | HS | 14.90 | 8.70 | 23.22 | |
| Slope of a linear fit to a log plot of the ISI values (maxspike) | OLM | 0.12 | 0.08 | 0.18 | **0.000001** |
| | HS | 0.26 | 0.19 | 0.33 | |
| Third inverse interspike interval (steady state) | OLM | 15.51 | 12.65 | 17.87 | **0.000003** |
| | HS | 25.19 | 18.92 | 29.15 | |
| Maximum voltage (maxspike) | OLM | 32.82 | 29.77 | 41.18 | **0.000004** |
| | HS | 24.54 | 17.05 | 27.47 | |
| Maximum voltage from voltage base (maxspike) | OLM | 98.90 | 95.47 | 104.17 | **0.000006** |
| | HS | 88.96 | 85.43 | 92.85 | |
| Time to second spike (maxspike) | OLM | 14.15 | 12.33 | 16.33 | **0.000007** |
| | HS | 9.20 | 7.95 | 10.95 | |
| Fourth inverse interspike interval (maxspike) | OLM | 101.53 | 95.75 | 103.36 | **0.000010** |
| | HS | 142.86 | 109.29 | 157.49 | |
| AP amplitude from voltage base (maxspike) | OLM | 92.06 | 89.10 | 94.85 | **0.000013** |
| | HS | 83.78 | 78.86 | 88.14 | |
| Peak of first AP (maxspike) | OLM | 32.23 | 29.44 | 41.18 | **0.000019** |
| | HS | 23.40 | 16.42 | 27.47 | |
| Maximum FI slope (global) | OLM | 0.31 | 0.31 | 0.38 | **0.000021** |
| | HS | 0.44 | 0.38 | 0.59 | |
| Second inverse interspike interval (steady state) | OLM | 20.73 | 14.28 | 22.92 | **0.000024** |
| | HS | 30.30 | 26.08 | 34.89 | |
| Single burst ratio (maxspike) | OLM | 0.67 | 0.56 | 0.82 | **0.000064** |
| | HS | 0.42 | 0.39 | 0.61 | |
| First inverse interspike interval (steady state) | OLM | 23.08 | 17.65 | 28.74 | **0.000072** |
| | HS | 36.90 | 28.50 | 49.64 | |
| Fast after hyperpolarization (steady state) | OLM | 21.83 | 19.18 | 22.87 | **0.000080** |
| | HS | 15.53 | 13.88 | 17.08 | |
| Average height of AP's (steady state) | OLM | 31.65 | 27.16 | 39.49 | **0.000133** |
| | HS | 22.57 | 13.85 | 26.10 | |
| Mean frequency (maxspike) | OLM | 69.90 | 65.89 | 84.23 | 0.404086 |
| | HS | 85.11 | 61.75 | 96.52 | |

*(Continued)*

**Table 4.** (Continued)

| Feature | OLM/HS | Median | Lower quartile | Upper quartile | P (Mann–Whitney U test) |
|---|---|---|---|---|---|
| Adaptation index—Normalized average difference of consecutive ISI's (maxspike) | OLM | 0.00 | 0.00 | 0.01 | *0.001094* |
| | HS | 0.01 | 0.00 | 0.01 | |
| Half width of the AP duration (maxspike) | OLM | 0.75 | 0.69 | 0.83 | 0.655148 |
| | HS | 0.76 | 0.71 | 0.81 | |
| Time to first spike (maxspike) | OLM | 4.80 | 3.68 | 5.80 | *0.009415* |
| | HS | 3.20 | 2.90 | 4.55 | |
| Fast after hyperpolarization (maxspike) | OLM | 17.36 | 14.42 | 19.51 | *0.001645* |
| | HS | 13.81 | 12.63 | 14.32 | |
| Adaptation index—Normalized average difference of consecutive ISI's (steady state) | OLM | 0.05 | 0.03 | 0.06 | 0.874394 |
| | HS | 0.05 | 0.03 | 0.07 | |
| Time to first spike (steady state) | OLM | 30.35 | 22.63 | 44.88 | 0.103920 |
| | HS | 24.20 | 21.25 | 29.70 | |
| Input resistance (standard negative) | OLM | 89.29 | 77.66 | 124.50 | *0.009415* |
| | HS | 129.41 | 98.39 | 141.11 | |
| Ratio between sag amplitude and maximal sag from voltage base (standard negative) | OLM | 0.42 | 0.32 | 0.47 | 0.215333 |
| | HS | 0.39 | 0.34 | 0.41 | |

OLM: $n = 24$, HS: $n = 23$, Underlined *p-values* show significant differences, italic *p-values* show significant diff. only without Bonferroni correction.

HS, hippocampo-septal; OLM, oriens-lacunosum-moleculare.

distributions and total predicted numbers of synapses for the 2 cell types, we estimated that, although the total length of the dendrites was similar (Table 1), HS cells have a significantly larger number of both excitatory and inhibitory synaptic inputs compared to OLM cells. OLM cells receive 7,186 (median) p-glutamatergic synapses (interquartile ranges: 5,919 to 8,538) and 1,261 p-GABAergic synapses (interquartile ranges: 1,015 to 1,603), whereas HS cells are innervated by 12,931 (interquartile ranges: 10,475 to 15,906) p-glutamatergic synapses and 2,654 (interquartile ranges: 2,152 to 3,255) p-GABAergic synapses, and 14.9% and 17% of the synaptic inputs of OLM and HS cells are p-GABAergic, respectively.

The majority of inputs on both cell types arrived onto the dendritic tree, while only 3% and 4% of all excitatory inputs and 3% and 7% of all GABAergic inputs targeted the cell body of OLM and HS cells, respectively. Most dendritic synapses were located on the shaft: 77% of all excitatory and 83% of all inhibitory synapses in OLM cells and 85% of all excitatory and 87% of all inhibitory synapses in HS cells; 20% of all excitatory and 14% of all inhibitory synapses innervated dendritic spines in OLM cells and 11% of all excitatory and 5% of all inhibitory synapses were on dendritic spines in HS cells.

In addition, a larger proportion of these inputs targeted proximal versus distal parts of the dendritic tree in HS neurons (skewness values were strictly positive for the distributions of both excitatory and inhibitory synapses in HS cells, and they were significantly larger than the skewness values in OLM cells; *T* test, excitatory synapses: $p = 0.0081$, inhibitory synapses: $p = 0.001$) (Fig 8).

## Modeling subthreshold responses in OLM and HS neurons: Model construction

Next, we constructed morphologically detailed but biophysically simplified functional models of OLM and HS neurons based on our experimental data. As our aim was to capture the subthreshold behavior of these cells, we included only those mechanisms that substantially

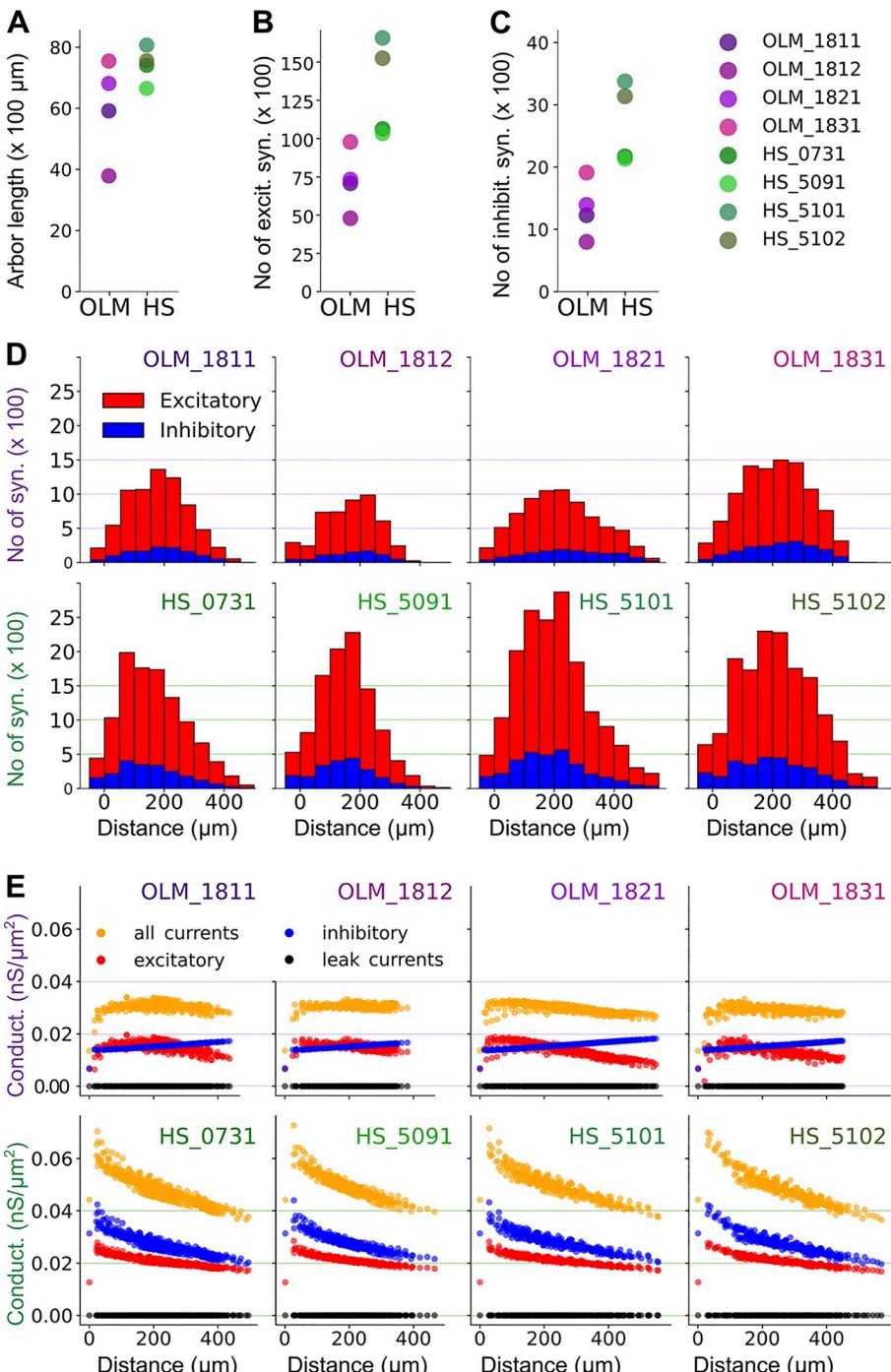

**Fig 8. Numbers and distribution of synaptic inputs in morphologically detailed models of OLM and HS neurons.**
(A) The total length of the dendrites in the morphological reconstructions. OLM = 5,917, 3,788, 6,816, 7,547 μm.
HS = 7,409, 6,654, 8,068, 7,558 μm. (B, C) The total number of excitatory synapses (panel B: OLM = 7,062, 4,776, 7,311,
9,766. HS = 10,620, 10,330, 16,570, 15,243) and inhibitory synapses (panel C: OLM = 1,229, 801, 1,293, 1,913.
HS = 2,172, 2,133, 3,375, 3,136) predicted for each model cell. In panels A–C, the circles represent the cells, and each
cell is shown in the same color throughout the panels. (D) The number of excitatory and inhibitory synapses in each
model as a function of the distance from the soma (bin size = 50 μm). See raw data in S23 Data. (E) The effective
conductance densities in each segment of the model neurons in the high-conductance state (see main text). Color red
marks the excitatory and blue the inhibitory current, black is the leak conductance and orange is the sum of the 3
(called "all currents" in this figure, although it excludes the voltage-gated current Ih, which is also implemented in the
model). Each circle represents a segment of the model at different distances from the soma. HS, hippocampo-septal;
OLM, oriens-lacunosum-moleculare.

influence the electric behavior of neurons in this membrane potential range. Specifically, our models contained, in addition to the membrane capacitance, the axial resistance (whose value was fixed to 100 Ohm cm), and the passive leak conductance, a model of the hyperpolarization-activated current (Ih). The model of Ih was based on the experimental results of [63] regarding the properties of Ih in OLM cells; we assumed that Ih had similar characteristics in HS cells, with possible variations in the amount of Ih as well as in the relative contribution of fast and slow components of Ih activation dynamics. Overall, for each of our model cells, we considered 5 unknown parameters: the specific membrane capacitance (cm), the conductance (gpas) and reversal potential (epas) of the leak current, the maximal conductance (gmax_H) of Ih, and the relative contribution of the fast component to the dynamics of Ih (fastRatio_H). All of these parameters were assumed to be spatially uniform. These unknown parameters were fitted to match the electrophysiological features extracted from voltage responses to negative current steps in the whole-cell recordings described above.

The best-fitting parameters for all models are listed in S13 Data. These covered similar ranges in the 2 cell types, although there was a tendency for higher conductance values in OLM cells compared to HS cells, especially for Ih. The resulting models were then validated using the Somatic Features Test of HippoUnit (S5 and S6 Figs), including features that were not used during parameter optimization, but still restricted to hyperpolarizing current injections. This validation confirmed that our models provided a good description of the subthreshold behavior of OLM and HS neurons at the level of the somatic voltage response to local current inputs, as the overwhelming majority of the model features were within 2 standard deviations of the corresponding experimental mean value (79/81 in OLM cell models and 72/81 in HS cell models).

## Modeling subthreshold responses to synaptic inputs in isolated neurons

To reveal the functional consequences of the morphological differences between the 2 cell types that were indicated by our EM analysis, we examined the responses of our model neurons to synaptic inputs arriving at different locations in the dendritic arbor. We first considered the effects of single synapses activated in isolation; this corresponds to in vitro experimental conditions with a low level of neuronal activity in the hippocampal network, which will be referred to as the "silent state." In the absence of relevant data regarding the properties of unitary synaptic currents in HS cells, we assumed that most parameters characterizing the primary effects of synaptic input (rise time, decay time, reversal potential) were the same in the 2 cell types (S14 Data) and corresponded to values measured in OLM cells [13,33,64]. However, as results from other cell types showed that the size of the synapse was strongly correlated with peak synaptic conductance (for a given type of synapse) [12,13,33,64] we took advantage of our EM data to estimate the relative strength of (excitatory and inhibitory) synapses in the 2 cell types. Since we found no significant difference in the size of excitatory synapses, we used the same maximal conductance value (calculated from measurements in OLM cells; [12]) in the 2 cell types. On the other hand, our EM data indicated that inhibitory synapses onto HS cell were, on average, 15% larger than those targeting OLM cells, and we therefore used a 15% larger value for the peak synaptic conductance in HS cells (S14 Data), which resulted in a 15% larger amplitude of (local) inhibitory synaptic currents compared to the experimentally measured value in OLM cells [13].

We activated synapses with these properties, which were assumed to be uniform within a given cell type, at different dendritic locations in our model neurons, and measured the amplitude of the somatic postsynaptic potential (PSP) using the Post-synaptic Potential Attenuation Test of HippoUnit. We found that, under such isolated conditions, both excitatory and

inhibitory synapses generated larger amplitude somatic PSPs in HS neurons than in OLM cells (Fig 9A and 9B). We also note that the amplitude of EPSPs and (especially) IPSPs showed only a moderate dependence on the distance of the synapse from the soma.

## Modeling subthreshold responses to synaptic inputs in the presence of background inputs

Neuronal populations in vivo are continuously active in a brain state-dependent manner. As a result, individual neurons in the hippocampal network are constantly bombarded by many excitatory and inhibitory synaptic inputs, which lead to a sustained increase in the membrane conductance (and a possible shift in the equilibrium potential) in all parts of the dendritic tree. These effects can greatly influence the processing of synaptic inputs within the cell, pushing individual neurons and the whole network into what has been referred to as the "high-conductance state" (HCS) [54,65].

To model the behavior of neurons in the HCS, one needs to take into account the total membrane conductance, including contributions from synaptic, leak, and voltage-gated conductances. This requires the calculation of the temporal average of the excitatory and inhibitory synaptic conductance per unit area of membrane everywhere in the dendritic tree [54]. For a specific type of input, this can be computed as the product of 3 quantities: the number of synapses per unit membrane area, the time integral of the conductance change evoked by the activation of a single synapse, and the average firing rate of the presynaptic neuronal population. If the synaptic conductance change is approximated well by a single decaying exponential function, the second term in this product is simply the maximum conductance change multiplied by the decay time constant. As a result, the mean contribution of a set of synaptic inputs to the total membrane conductance can be calculated as follows [54]:

Conductance per area = synapse number per area * rate * peak conductance * tau

The density of synapses in any given section of our model neurons can be estimated from our EM data as described above (Fig 8). The peak conductance and the decay time constant (tau) for excitatory and inhibitory synapses were based on published values for OLM cells, and parameters for HS cells were assumed to be the same, except for a 15% larger value of the peak conductance of inhibitory synapses due to their larger size as discussed above. Finally, the firing rates of excitatory and inhibitory presynaptic populations can vary considerably (e.g., in different brain states), so we will use a particular set of plausible values for demonstration purposes (3 Hz for excitatory and 30 Hz for inhibitory afferents).

Fig 8E shows the contributions of the excitatory synaptic, inhibitory synaptic, and leak conductance to the total conductance density (excluding Ih, which is voltage dependent) in various parts of our model neurons. Under the conditions of the HCS as defined above, excitatory and inhibitory synapses are predicted to make contributions of comparable magnitude to the total membrane conductance, and both of these are several orders of magnitude larger than the contribution of the leak conductance (and Ih). Different models of the same cell type showed very similar conductance distributions. The total conductance was significantly larger in HS cells than in OLM cells, and it was almost constant at all locations in OLM cells, while it decreased substantially with increasing distance from the soma in HS neurons. The difference between the 2 cell types was due to the different dependence of synapse density on cellular parameters (such as distance from the soma).

Another consequence of the different contributions of excitatory and inhibitory synapses to the total membrane conductance in the different cell types is a difference in the local equilibrium potential of the cell membrane. This can be calculated as the mean of the reversal potentials of all ionic currents weighted by their respective conductances. We calculated the local

equilibrium potential at every segment of the models and observed a clear difference between the 2 cell types. The proximal dendrites of the OLM cells had more depolarized equilibrium potentials than the proximal dendrites of the HS cells but became gradually more hyperpolarized further away from the soma, while the dendrites of the HS cells had a more uniform distribution of dendritic equilibrium potentials with slightly more depolarized values further away from the soma (S7 Fig). These differences affect the steady-state membrane potential distribution within the cells and the driving force associated with excitatory and inhibitory synaptic inputs at different dendritic locations.

Using these calculated effective values of the tonic membrane conductance (including the contributions of synaptic conductances as well as the leak conductance), we repeated our in silico experiments that measured the somatic effects of activating a single (excitatory or inhibitory) synapse at different locations in the dendritic tree (Fig 9C and 9D). The results showed that the inclusion of tonic background activity fundamentally changed the effects of single synaptic inputs in both cell types. In the HCS, even proximal synapses evoked at least one order of magnitude smaller PSPs than in the silent state, and the magnitude of the predicted response became almost negligible beyond approximately 200 micrometers from the soma, so that the majority of single synapses evoked very small somatic voltage responses. In addition, because of the larger attenuation of synaptic signals in HS cells due to their higher tonic membrane conductance, single somatic PSPs at a given distance from the soma were smaller in HS cells than in OLM cells in the HCS, in contrast to what we observed in the silent state. On the other hand, the distributions of PSP amplitudes in the 2 cell types became strongly overlapping, indicating that the effects of single synaptic inputs could be quite similar on average, although the strongest synapses were predicted to be more potent in OLM cells in this state.

Finally, we investigated whether taking into account the presence of mitochondria has a significant effect on signal propagation in the models. To do this, we first observe that, from the perspective of electrical signal propagation, the main effect of the presence of mitochondria in dendrites is to decrease the effective cross section that is available for axial current flow. In our models, we can compensate for this effect by increasing the specific axial resistance. More specifically, we multiplied the specific axial resistance by the ratio of the total cross-section of the dendrite and the cross-section excluding mitochondria (proportion of mitochondria in Table 1 and S2 and S3 Data files). Since we did not have data on the exact location of mitochondria in our reconstructed morphologies, we applied the correction based on a modified axial resistance uniformly in the dendrites, using the ratio of the total dendritic volume and the volume without mitochondria in our samples of EM data from the 2 cell types. Using this method, we did not see any significant change in the behavior of our model neurons, suggesting that mitochondria do not have a significant effect on signal propagation in the dendrites of OLM and HS neurons.

## Modeling spiking responses in OLM and HS neurons: Model construction

As presented above, our anatomical and physiological data allowed us to construct morphologically detailed, essentially passive models of OLM and HS cells, and to investigate dendritic signal propagation and synaptic integration in the subthreshold voltage range. A natural next question is how action potentials are generated in these neurons, and how their spike output is influenced by the spatial and temporal pattern of the synaptic inputs that they receive. These questions can be addressed effectively in fully active, morphologically and biophysically detailed neuronal models. The construction of such models is also facilitated by our anatomical and physiological data; however, building such an active model also requires knowledge of the identity, quantity, and spatial distribution of voltage-gated ion channels in the cell

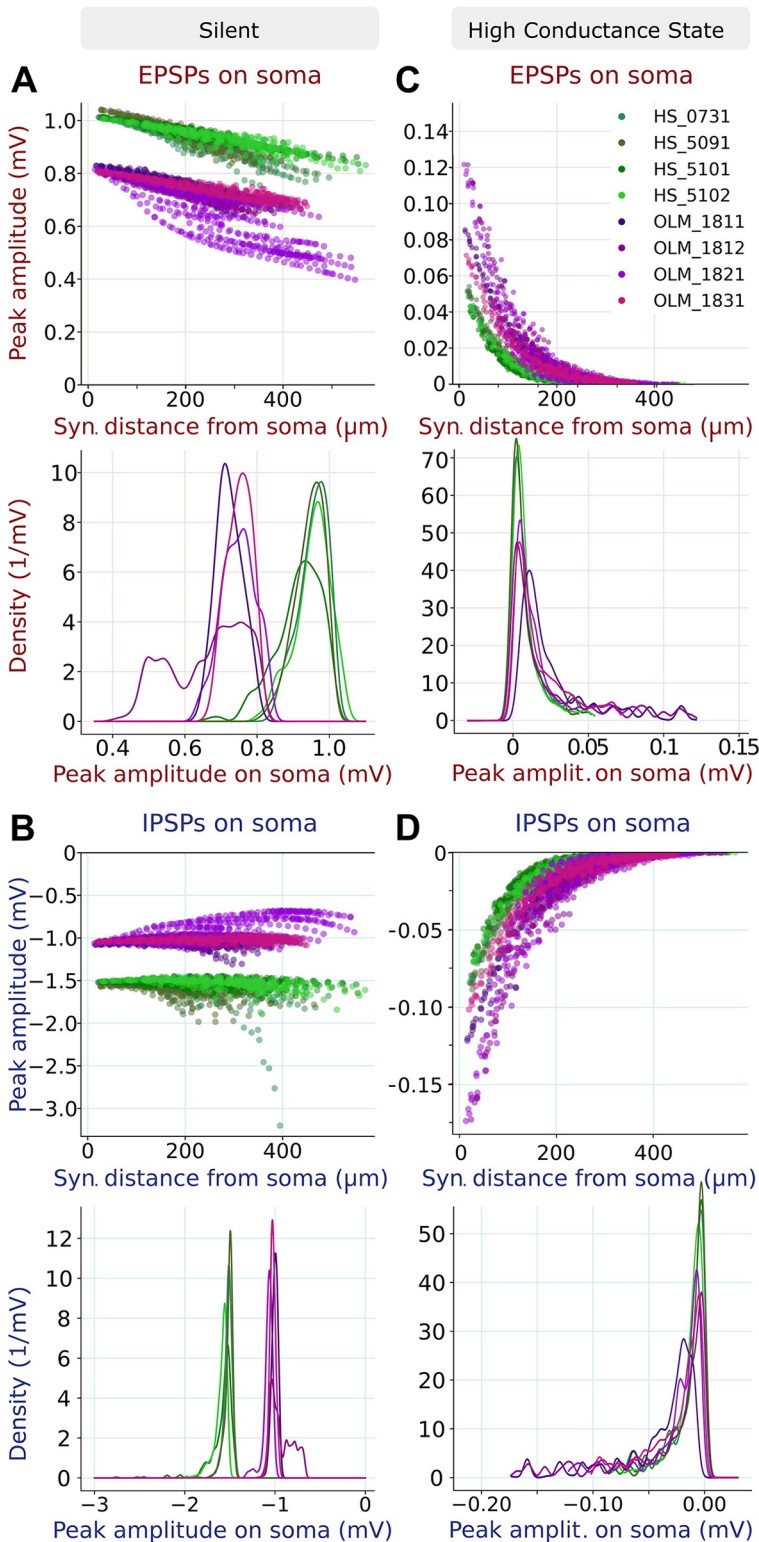

**Fig 9. Analysis of simulated excitatory and inhibitory synaptic potentials in models of OLM and HS cells.** (A, B) (top) The peak amplitude of the EPSPs (A) and IPSPs (B) measured at the soma in response to synaptic inputs to segments at various distances from the somata of the 4 OLM and 4 HS cell models, simulated in the silent state. Each color represents 1 cell, and each circle is a different dendritic segment. In the bottom panels, we show the density distributions of the somatic PSP amplitudes. Panels C and D show the results of the same in silico experiment in the high-conductance state. HS, hippocampo-septal; OLM, oriens-lacunosum-moleculare; PSP, postsynaptic potential.

membrane. Some (but certainly not all) of the required parameters have been measured in OLM cells [66–68], and this has enabled the construction of detailed multi-compartmental models of OLM cells [68–72], while there is practically no information available about voltage-gated channels in HS neurons.

Nevertheless, our in vitro physiological data from the 2 cell types indicated that they showed relatively similar behavior (at least in terms of the somatic voltage response to somatic step current injections), which encouraged us to build and simulate a set of prototype active models of both cell types based on our data using the same set of biophysical mechanisms. We used the same reconstructed morphologies as above (4 OLM and 4 HS cells), and took the models of ionic conductances and the basic patterns of their spatial distributions from a previously published detailed model of OLM cells [68]. We tuned the maximal conductance values of the voltage-gated channels and leak channels (14 parameters in total) via Neuroptimus using a feature-based error function as before, but this time including features extracted from responses to both depolarizing and hyperpolarizing current injections recorded in the appropriate cell type.

The best-fitting parameters for all models are listed in S16 Data. The resulting models were then validated using the Somatic Features Test of HippoUnit (S8 and S9 Figs), including features that were not used during parameter optimization. The results of this validation indicated that the active models captured the physiological behavior of OLM cells reasonably well (335 to 361 out of 447, i.e., 75% to 81% of features were within 2 standard deviations of the experimental mean). Considering only the features that describe the intensity and pattern of spiking (these include spike numbers, spike delays, and interspike intervals), the match between the model output and the experimental results was even better (90 to 100 out of 100 features within 2 standard deviations). The responses of HS cell models matched the corresponding experimental data less accurately (237 to 285/505, or 47% to 56% of all features and 62 to 69/106, or 58% to 65% of spike pattern features within 2 standard deviations). These results suggest that the set of biophysical mechanisms that determine the physiological behavior of the 2 cell types may not be identical, and achieving a more accurate description of HS cell physiology will require direct measurements of voltage-gated channel properties and distributions in this cell type.

## Modeling the responses of active neurons to synaptic inputs

Our active models of OLM and HS neurons, combined with our data on synaptic distributions, also allow the investigation of spiking responses in the 2 cell types in response to various spatial and temporal patterns of synaptic input. While a complete analysis of all the relevant scenarios is beyond the scope of the current study, and may also require further experimental data and the improvement of the active neuronal models, we demonstrate the feasibility of this approach by showing the results of 2 sets of simulations.

First, we determined how the output firing rate of the model neurons depended on the number and location of active synaptic inputs. In the simplest case, we activated a variable number of excitatory synaptic inputs selected randomly according to the predicted distributions of synapses on the dendrites of individual reconstructed neurons. Synaptic inputs were activated independently and randomly at a constant rate of 10 Hz. We found, as expected, that a certain number of excitatory synapses had to be active for the neurons to start generating action potentials, and the firing rate increased monotonically as the number of active inputs was increased further (S10 Fig). We also repeated the same simulation experiments targeting only proximal (0 to 100 μm from the soma) or distal (300 to 400 μm from the soma) synapses. The qualitative picture was similar to the baseline case in both of these situations, but we found that a smaller number of proximal synapses was required to drive spiking in most cells

compared to distal or widely distributed ones (S10 Fig). Finally, we used the estimates of synapse numbers that we had derived by combining the morphologies with the synaptic densities measured by EM to plot the output spike rates as a function of the percentage (rather than the absolute number) of activated synapses in the given cell and distance range (S10D–S10F Fig). These percentages are expected to provide a better measure of the level of activity in the relevant presynaptic populations, independent of the morphology of the target neuron. There were substantial differences between the individual models in the threshold percentage (typically less than 5%) and in the steepness of the activation curve, but there was no systematic difference between models of OLM and HS cells in these characteristics.

Second, we investigated whether the spike output of the model neurons could be modulated by synaptic input whose rate varied in a sinusoidal manner at theta frequency (7 Hz). We stimulated 5% of the excitatory synapses randomly distributed on the whole dendritic arbor, activating synapses independently at a rate which varied between 1 and 21 Hz during the theta cycle. This was meant to represent inputs from local pyramidal neurons during behavioral states characterized by theta oscillations in the population activity. Then, we calculated the spiking rate of our active OLM and HS model neurons at each phase of the sinusoid to evaluate theta modulation (S11 Fig). We used Rayleigh's Test of Uniformity to detect significant phase dependence and measured the mean vector length to determine the depth of modulation. Each of the 4 HS cells showed significant theta modulation ($p < 0.001$ according to Rayleigh's test; mean vector lengths: 0.467 to 0.642) and so did 3 of the 4 OLM cells ($p < 0.001$; mean vector lengths: 0.2 to 0.676). The fourth OLM cell fired only a few spikes at this level of synaptic stimulation; however, when we repeated the simulation with 15% of excitatory synapses activated, this model also showed significant theta modulation ($p < 0.001$; mean vector length: 0.36). Overall, no clear difference was found in the degree of modulation by theta frequency inputs between the 2 cell types.

## Discussion

Understanding how the brain implements various cognitive functions requires detailed knowledge of the properties of the different classes of its constituent neurons and their synaptic interactions [73,74]. The commonly used abstraction of the point-like neuron in computational models completely ignores the powerful computational properties of dendrites that allow the isolation of synaptic inputs that enable multiple parallel operations [75]. Recently, large-scale, biologically realistic data-driven models, in particular, have emerged as powerful tools in the investigation of brain circuits [76–84], but they explicitly rely on detailed information about neuronal morphologies, electrophysiological characteristics, connectivity, and synaptic properties, and the quality of these data is a crucial determinant of the faithfulness and predictive power of such models.

### OLM and HS cell types in hippocampus CA1

OLM and HS cells are hippocampal somatostatin-positive GABAergic neurons in the CA1 area that have similar dendritic morphology (at least at the light microscopic level) but different axonal projections, firing patterns, and functional roles in network activity [8,26]. To characterize the anatomical properties (including inputs and outputs) of the 2 cell types in more detail, we first reconstructed their dendritic trees in 3d at the light microscopic level ($n = 4–4$ cells) and then we created a database of their 3-dimensionally reconstructed dendritic segments, cell bodies, and axonal segments including their input and output synapses ($\Sigma\ n = 90$ dendritic segments, 18 branching points, 14 somata, 24 axonal segments, 5,844 input, and 227 output synapses). We measured several anatomical parameters that are important in the electrical conductance of cell

membranes, e.g., the surface area of dendritic segments, the volume of cytosol and mitochondria, and the density, number and size of excitatory/inhibitory inputs. All data were corrected for volume changes during the sample processing. These parameters were precisely incorporated into the morphologically detailed compartmental model neurons we reconstructed to examine how the input organization influences signal propagation in the 2 cell types.

### Dendritic morphologies

We found that in the case of OLM and HS cells, the thickness of the dendritic segments is highly variable. Since we also found that the thicker dendrites receive proportionally more inputs, the linear density of inputs is also highly variable along the dendritic tree and it could not be precisely predicted without knowing the thickness of the dendritic segment. This highlights the importance of accurately tracking dendritic thickness during whole cell reconstructions, if they are used to estimate the number of synaptic inputs.

### Comparisons of fine ultrastructural properties of OLM and HS cells

Comparing the anatomical parameters of reconstructed dendritic segments of OLM and HS cells, we found several significant differences (Fig 3D and 3E). The numerical density of both p-glutamatergic and p-GABAergic synaptic inputs (and consequently the synaptic coverage on the plasma membrane) was significantly higher on HS cells. Most of the HS cell dendritic segments were almost completely covered by converging afferent terminals (see Fig 3B and 3C), whereas larger terminal-free membrane regions were detected on OLM dendrites (Fig 3A and 3C). In an earlier study, using serial transmission electron microscopy, we estimated the density of GABA-negative and positive inputs of HS cells in rat [44]. Similarly to mouse HS cells, they were completely covered by terminals and compared to other examined interneurons in the CA1 area of rat hippocampus they received the most abundant synaptic inputs [54,85,86]. Here, the ratio (%) of GABAergic inputs per all inputs was similar in the case of mouse OLM and HS cells (14.3% and 15.4%, respectively, this study), and these were similar in rat HS cells (14%) [44]. This suggests that more excitatory inputs require proportionately more inhibition, although very different GABAergic percentages have been also detected in other cell types [54]. The higher number of synaptic inputs likely means that HS cells sample the average activity of large populations of PCs more efficiently than OLM cells. HS cells seem to function like antennae or probes that keep broadcasting the activity state of the hippocampus CA1 to the MS (and probably to the subiculum). Conversely, OLM cells are more selectively involved in local computations, including the selection of engram cells, which may require fewer inputs with a more specific distribution from subcortical afferents and local PCs.

More synapses were associated with significantly more mitochondria in HS cell dendrites compared to OLM cell dendrites. This finding explains why HS cells express a higher than average level of cytochrome C (a central component of the oxidative phosphorylation chain) compared to other somatostatin neurons in str. oriens [87]. This may suggest a higher metabolic rate that may be explained by the large number of inputs that they need to integrate, while they must also maintain the large axonal tree far away from their somata. Indeed, during sharp wave ripples double-projection (HS) cells increase their firing rates [36] demanding more intense ATP production, whereas OLM cells are inhibited during these events [49].

### Source of afferent synapses

Since many anatomical data are only available from rats, we have to rely on these earlier studies when discussing the source of the inputs to OLM and HS cells in the mouse. However, the anatomical organization is very similar in the 2 species [85].

The dendritic trees of OLM and HS cells arborize in the termination zone of local axons of CA1 PCs in str. oriens/alveus [88] and were shown to receive the majority (at least 75% to 83%) of their p-glutamatergic inputs from CA1 PCs [28]. The str. oriens of CA1 receives a small number of entorhinal axons as well (alvear pathway) [89], which preferentially innervate GABAergic cells [61]. Therefore, these entorhinal axons could also be among the p-glutamatergic inputs of OLM and HS cells. Schaffer collaterals of CA3 PCs likely provide input to OLM and HS cells only very rarely, as they innervate PCs to a much greater extent than interneurons [61,90]. Based on the above, the source of glutamatergic innervation of the 2 cell types is probably similar, although HS cells receive substantially more inputs.

The source of the p-GABAergic inputs is probably more diverse and more consequential, because both cell types may receive inputs from a variety of local interneurons and extrinsic GABAergic afferents. Both OLM and HS cells are among the targets of GABAergic medial septal inputs [13,44,91–93]. Relaxin-containing axons from the nucleus incertus also provide an important external GABAergic input to both cells types [24]. From the possible local inputs, type III interneuron-selective interneurons were shown to be particularly important for both cell types [13,94–96]. GABAergic inputs arriving from different sources can be very different in terms of their synapse size, temporal dynamics, efficiency, etc. [13,97]. Furthermore, they can preferentially innervate certain cell types or distinct membrane domains of these cells (soma, proximal, or distal dendritic segments) [5]. We hypothesize therefore, that the differences in the sizes of GABAergic synapses on OLM and HS somata and dendrites (and between somata and dendrites within the same cell type) are due to this variability and reflect the fact that the 2 cell types (and their different membrane domains) are probably differently innervated by inputs from different sources. This may also play a role in their distinct behaviors during different network activity patterns [36,49].

In addition to typical GABAergic cells, MS cholinergic cells are also GABAergic and all of their terminals establish gephyrin-positive p-GABAergic inputs in CA1 [98]. OLM cells were reported to receive cholinergic inputs which were essential for their activation during memory process [21,34]. Although cholinergic synapses are smaller than the non-cholinergic GABAergic synapses, the GABAergic synapses did not show bimodal size distribution.

## Correlation of synapse size with the distance from soma

We found that the sizes of p-glutamatergic inputs increased towards the distal dendrite on OLM cells, whereas the sizes of HS synapses showed an opposite effect. Similar trends were found in the case of GABAergic inputs, although they did not show significant correlations probably because of their smaller numbers. Generally, larger synapses contain proportionally more postsynaptic receptors [99] and elicit larger synaptic events. The increase in synapse size at distal dendrites can make these distal synapses more efficient and can compensate for the electrotonic filtering of inputs to distal dendrites in OLM cells. According to our modeling results, this would make all synapses (of the same type) about equally efficient in terms of the amplitude of the somatic PSP, at least in an isolated neuron. On the other hand, our simulations indicate that such a moderate increase in the synaptic conductance would fail to offset the much greater attenuation of distal inputs in the presence of a large amount of background synaptic activity in active network states. Further research is needed to understand why the opposite strategy is adopted by HS cell dendrites and what the consequences for synaptic integration might be. A detailed modeling approach like the one used here could be particularly useful in this context; however, to investigate and compare different strategies for dendritic integration, the model may need to be extended to incorporate active dendritic processes (which are known to be present, at least in OLM cells [66]), as demonstrated previously for CA1 pyramidal neurons [100].

## OLM and HS cells preferentially target pyramidal cells

OLM cells are local interneurons with dense axonal trees that primarily target the str. lacunosum-moleculare [27,101,102]. In contrast, HS cells are long-range projecting cells that send axons to the MS and probably all of them also to the subiculum, but also have sparse local axonal branches in the hippocampus [36,45]. Local axons of mouse HS cells reconstructed in this study arborized in str. oriens and radiatum and never entered str. lacunosum-moleculare. Compared with the dense axonal tree of OLM cells, HS cells formed sparse branches extending far from the soma, similar to rat HS cells [36,44]. The axonal tree of our best filled biocytin-labeled OLM cell was 22,780 μm long in a 300 μm thick slice (Neurolucida reconstruction, Fig 7). Multiplying this number by linear density of 0.404 synapses/μm (data from our EM reconstructions) gives 9,203 synapses. The axonal branches of this cell extended to 818 μm in the medio-lateral direction. We can assume that the axon tree also extends at least this far in the rostro-caudal direction. If we assume that we reconstructed only a 300 μm thick slice of the 818 μm diameter disc-like axon arbor field, then we can estimate that this OLM cell established approximately 16,900 local synapses (depending on the position of the slice in the disc-like arbor). Because inter-varicose segments of a mouse OLM axon also form synapses and an axon bouton can sometimes make 2 synapses (see Fig 5B), it is hard to compare this numbers with earlier investigations where only the total number of boutons (16,874), but not the synapses of an OLM cell was estimated, especially because this number was calculated in rat [101].

Here, we reconstructed the local axonal tree of BDA-labeled HS cells as well. Because the DAB labeling of local axons faded away from the soma towards the distal branches and we could analyze only a limited number of sections, these are considered only partial reconstructions. The best filled HS axonal arbor was 8,625 μm long locally (only 8 pieces of 60 μm thick sections, i.e., a 480 μm thick block was examined in the rostro-caudal direction). If we multiply this length by the linear density of 0.45 synapses/μm (data from our EM reconstructions), it gives a total of 3,881 local synapses in that 480 μm thick block. However, previously in the rat we found that bouton-bearing collaterals of HS cells spread all over more than half of the entire dorsal CA1 rostro-caudally, and in the entire medio-lateral extent of CA1 [44], some cells reached even the CA3 area. The dorsal mouse CA1 is about 3.5 mm. Therefore, if these mouse HS cells also arborize in about half (1.75 mm) of the entire dorsal CA1, then HS cells may have as many as about 14,149 synapses locally in the CA1 (and they may have further synapses in CA3 as well). Therefore, HS cells may have just as many local synapses as other classical local interneurons but probably reach much more PCs (with fewer synapses per PC) in the hippocampus, while they target extra-hippocampal areas as well.

Although HS cells were found to target interneurons in in vitro slice preparations of young rats [45], both cells are known to innervate mainly PCs and to a lesser extent (3.3% to 14%) interneurons in adult rats [29,30,37,44]. We found that OLM and HS synapses primarily targeted PCs (97%, 96%, respectively) and often targeted their spines (19%, 23%, respectively) that is especially efficient to modulate excitatory inputs of the temporo-ammonic pathway (OLM cells) or Schaffer collaterals (HS cells) on the same spines [11]. However, OLM cells established several orders of magnitudes higher number of synapses locally.

We also found that only 1% of OLM cell synapses and 2% of HS cell synapses targeted GABAergic interneurons. Because the ratio of GABAergic neurons is about 9.9% in the CA1 area of the dorsal hippocampus [103], this suggests that these cells preferentially innervate PCs.

The sizes of axonal output synapses of OLM axons were significantly larger than those of HS cells. In addition, the volumes of individual mitochondria were also larger in OLM axons. OLM axons innervate more distant dendritic segments of CA1 PCs in str. lacunosum-

moleculare than HS cells in str. radiatum and oriens, and they may form larger synapses to compensate for the larger electrotonic distance.

## OLM and HS cells proportions of SOM-positive interneurons in CA1 str. oriens

In CA1 str. oriens, there are other subsets of somatostatin-positive interneurons in addition to OLM and HS cells (bistratified cells, back-projection neurons, oriens-retrohippocampal projection cells) [8]. We estimated the density of HS and OLM cells and found that they account for at least 49.1% and 9.5% of CA1 str. oriens somatostatin-positive cells, respectively. Nichol and colleagues found the same proportion for OLM cells (47%) [104], whereas Jinno and Kosaka found a larger population (22.5%) of HS cells using a different tracer [43]. It should be noted that these values must be regarded as minimum values since probably not all cells could be labeled using tracing methods.

## Electrophysiological characteristics of OLM and HS cells

Responses of the 2 cell types to sustained somatic current injections were generally similar, with some potentially important differences. Specifically, HS neurons showed a clear propensity to generate high-frequency bursts following the onset of intense stimulation, which may contribute to their high activity during sharp wave-ripples.

## Functional considerations

Our results allow us to zoom in on the possible reasons for the distinct firing activity of OLM and HS cells in vivo, and particularly during sharp wave-ripple (SWR) activity. The relatively minor differences in intrinsic physiological features that we found are unlikely to fully account for the in vivo findings. Therefore, the difference in the level of SWR-related activity (high in HS cells and very low in OLM cells) probably comes from differences in the amount of excitatory and/or inhibitory input that they receive. We found that HS cells received a significantly larger number of both excitatory and inhibitory synaptic inputs than OLM cells; however, the ratio of excitatory and inhibitory synapses was similar in the 2 cell types. In addition, our modeling results indicated that the larger number of active synaptic inputs in HS cells would increase the membrane conductance more than in OLM cells, thereby effectively shunting further synaptic inputs and thus reducing the functional effect of the difference in synapse number. The typical sizes of synapses were also similar in the 2 cell populations, the only significant difference being the slightly larger size of inhibitory synapses in HS cells, which, if we take synapse size as a proxy of synaptic strength, would actually predict larger inhibition and thus lower activity in HS cells during SWRs, which is the opposite of the experimental observations. Overall, the observed differences in synapse numbers and sizes are also unlikely to account for the distinct firing activity during SWRs.

Therefore, the most likely remaining explanation is that the responses of OLM and HS cells are dominated by excitatory and/or inhibitory input from different sources, with very different activity patterns during SWRs. This is interesting in itself since the dendrites of the 2 cell types lie in the same layer of the hippocampus, and thus any difference in the sources of synaptic input would indicate selective innervation of one or both cell types. In fact, as we discussed above, there is some anatomical evidence for such selectivity, especially for inhibitory inputs.

Because most excitatory inputs come from local PCs in both cell types, the larger number of excitatory inputs in HS cells may contribute to higher activity during SWRs, when the average activity of local PCs is high. However, the fact that OLM cells are silent during these high-activity periods while HS cells fire at high rates indicates that OLM cells probably receive more

active inhibition during SWR activity, which also points to a (partially) different origin of the inhibitory synapses that they receive.

On the other hand, a significant role of other mechanisms, such as differences in the short-term plasticity of the inputs of OLM and HS cells, or differences in voltage-gated currents, cannot be ruled out, and would constitute exciting targets for future experimental and modeling research.

## Further modeling opportunities

Our morphological reconstructions and electrophysiological recordings enabled us to construct morphologically detailed models of OLM and HS cells at 2 different levels of biophysical realism. One set of models was designed for the investigation of subthreshold phenomena and included only Ih as a voltage-gated ion channel. The other set of models aimed to capture both subthreshold and suprathreshold (spiking) responses and included a wide variety of voltage-dependent biophysical mechanisms. Our primary goal was to demonstrate how the various data sets resulting from our experiments can be combined to enhance the quality of data-driven models of specific cell types. However, even with the simpler passive models and restricting our attention to the subthreshold voltage range, we were able to make some interesting predictions. For example, our simulations of intracellular responses to synaptic inputs predicted that, although synapses in all parts of the dendritic tree had similar effects on the somatic membrane potential in isolated cells (such as those recorded in most in vitro experiments), only the more proximal synapses had a measurable effect on somatic voltage during in vivo-like conditions with a high level of network activity.

Including the responses to positive current injections in the physiological target data provides additional constraints for the construction of fully active spiking model neurons. However, the spiking behavior of neurons depends strongly on the amounts and exact biophysical properties of the voltage-gated channels in their cell membranes. While there are some experimental findings available on voltage-gated ionic currents in OLM cells [66–68], there are (to our knowledge) essentially no such data published for HS cells. Therefore, in our models we used the same types of conductances and spatial distributions for HS cells as for the OLM cells. We used the previously published OLM cell model of Lawrence and colleagues [68] as a starting point and fitted the maximal conductances of its channels separately for each morphology to match the features derived from responses to depolarizing and hyperpolarizing currents in the corresponding cell type. It became quite clear from the parameter fitting and even clearer after the validation process that the models constructed in this way cannot entirely describe the active properties of the HS cells while they provide a good fit to the experimental data from OLM cells. To create better models for HS interneurons, more data would be required about their ion channel composition, the properties and localization of their ionic currents, and more information about their dendritic behavior. We also found that the models of both cell types (but especially OLM cells) were better at describing the spike rates and patterns of real neurons than at capturing finer details of the voltage response (e.g., action potential shape). The good accuracy of the models at predicting spikes allowed us to investigate the effects of synaptic inputs on the spiking output of these cells and suggests that the models may be suitable for inclusion in biophysically detailed network models.

It is important to note that in the case of the active models and their responses to synaptic stimulation, the variance within a cell type was greater than the difference between cell types. Increasing the number of cells in each population along with building HS models that are closer to the experimental observations could emphasize the differences of the 2 interneuron types. For now, we may conclude that since all the cells within 1 class were fitted to the same

electrophysiological data using the same conductances, morphological differences provide the most likely explanation of the variability of the cell models within the same cell type.

Finally, the data collected in this study also provides essential constraints for data-driven models at the network level. In particular, our results on the densities and numbers of input and output synapses of OLM and HS cells provide information on the connectivity of these cell populations in the hippocampal network and can be used during the reconstruction and validation of the connectome of area CA1 and its afferent and efferent brain regions [77,105,106].

## Methods

### Ethics statements

All experiments were performed in accordance with the Institutional Ethical Codex, Hungarian Act of Animal Care and Experimentation (1998, XXVIII, section 243/1998) and the European Union guidelines (directive 2010/63/EU), and with the approval of the Institutional Animal Care and Use Committee of the Institute of Experimental Medicine of the Hungarian Academy of Sciences (PE/EA/107-5/2021). All efforts were made to minimize potential pain or suffering and to reduce the number of animals used.

### Animals and surgery

A total of 13 C57Bl/6NTac mice (young adult, from both sexes), 15 nicotinic acetylcholine receptor alpha 2 subunit (Chrna2)-cre mice (young adult, from both sex, heterozygous/homozygous), and 1 SOM-Cre mouse (young adult female, heterozygous) were used in the present study. Mice were anaesthetized with isoflurane followed by an intraperitoneal injection of an anesthetic mixture (containing 8.3 mg/ml ketamine, 1.7 mg/ml xylazine-hydrochloride in 0.9% saline, 10 ml/kg bodyweight) and then were mounted in a stereotaxic frame.

For selective visualization of HS cells, we used 3 different retrograde labeling methods. See Fig 1 for detailed workflow. For reconstruction of dendritic trees and axons of HS cells, we injected 50 or $2 \times 50$ nl biotinylated dextrane amine (BDA-3000, Invitrogen) into the medial septal area (MS) of C57Bl/6NTac mice ($n = 6$). For the colocalization experiments (Fig 1), we injected yellow-green FluoSpheres (Thermo Fisher Scientific) into the MS of 2 C57Bl/6NTac mice or AAVrg-syn-FLEX-jGCaMP8m-WPRE (Addgene, 162378) into the MS of a heterozygous SOM-Cre female mouse. For the in vitro experiments, we used FluoSpheres-injected mice (see below). The coordinates for the injections were based on the stereotaxic atlas [107]: 1 mm anterior from the bregma, in the midline, and 4.5 and/or 4.9 mm below the level of the horizontal plane defined by the bregma and the lambda (zero level).

For selective labeling of OLM cells that are Chrna2-positive in the str. oriens, we injected 20 to 60 nl rAAV2/5-EF1a-DIO-eYFP tracer virus (UNC Vector Core; 4.4–8.5 × 1,012 colony forming units/ml) into the CA1 area of the hippocampus bilaterally into Chrna2-Cre mice. The coordinates for these injections were: −2.0 posterior from the bregma, +-1.5 or 1.75 laterally, and 1.4 mm below the zero level.

For the injections, we used a Nanoject 2010 precision microinjector pump (WPI, Sarasota, FL 34240) and borosilicate micropipettes (Drummond, Broomall, PA) with tips broken to 40 to 50 μm. After the surgeries, the animals received 0.5 to 0.7 ml saline for rehydration and 0.03 to 0.05 mg/kg meloxicam as a nonsteroidal anti-inflammatory drug (Metacam, Boehringer Ingelheim, Germany) intraperitoneally to support recovery, and we placed them into separate cages for 7 to 14 days (BDA and FluoSpheres-injected mice) or 23 to 39 days (virus-injected mice) before perfusions.

## Perfusions and sectioning

For perfusion, mice were deeply anaesthetized as above. Mice used for electron microscopic analysis of dendrites and somata were perfused transcardially first with 0.9% NaCl in 0.1 M phosphate buffer solution (PBS, pH = 7.4) for 30 s followed by a fixative containing 2% para-formaldehyde (PFA), 1% or 0.5% glutaraldehyde (GA), and 15% (v/v) picric acid in 0.1 M phosphate buffer (pH = 7.4; PB) for 5 min. Then, the brain was removed from the skull, cut sagittally and coronally into 4 pieces, and immersion fixed in a fixative containing 4% PFA and 0.2% to 0.25% GA in PB for 2 to 4 h at room temperature, on a shaker, then the fixative was washed out with PB.

Mice used for Neurolucida drawing of whole dendritic trees of OLM and HS cells and electron microscopic analysis of HS cell axons were perfused similarly (Fig 1), except that the fixative used for immersion did not contain glutaraldehyde. Mice used for somatostatin and eYFP/FluoSpheres double labeling were perfused transcardially first with PBS for 30 s followed by a fixative containing 4% PFA for 40 min, followed by PB for 10 min.

For the correction of shrinkage/dilatation of the tissue (see below), the volume of brains used for Neurolucida or electron microscopic reconstruction of cells was measured, then fore-brain blocks were serially sectioned using a Leica VT1200S vibratome at 60 μm. Before further processing, all sections were mounted on slides in PB and photographed using a Zeiss Axio-plan2 microscope. For measurement of shrinkage/dilatation during processing, see methods below.

## In vitro slice preparation

For acute slice experiments, FluoSpheres (HS cells, $n$ = 4 mice) or rAAV2/5-EF1a-DIO-eYFP tracer virus-injected mice (OLM cells, $n$ = 4 mice) were decapitated under deep isoflurane anesthesia. The brain was removed and placed into an ice-cold cutting solution, which had been bubbled with 95% $O_2$/5% $CO_2$ (carbogen gas) for at least 30 min before use. The cutting solution contained the following (in mM): 205 sucrose, 2.5 KCl, 26 $NaHCO_3$, 0.5 $CaCl_2$, 5 $MgCl_2$, 1.25 $NaH_2PO_4$, and 10 glucose. Then, coronal slices of 300 μm thickness were cut using a Vibratome (Leica VT1000S). After acute slice preparation, slices were placed into an interface-type holding chamber for recovery. This chamber contained standard ACSF at 35°C that gradually cooled down to room temperature. The ACSF solution contained the following (in mM): 126 NaCl, 2.5 KCl, 26 $NaHCO_3$, 2 $CaCl_2$, 2 $MgCl_2$, 1.25 $NaH_2PO_4$, and 10 glucose saturated with carbogen gas. All salts were obtained from Sigma-Aldrich or Molar Chemicals KFT.

## Intracellular recordings

After incubation, slices were transferred individually into a submerged-type recording chamber with a superfusion system allowing constantly bubbled (95% $O_2$–5% $CO_2$) ACSF to flow at a rate of 3 to 3.5 ml/min. The ACSF was adjusted to 300 to 305 mOsm and was constantly saturated with 95% $O_2$–5% $CO_2$ during measurements. All measurements were carried out at 33 to 34°C, the temperature of ACSF solution was maintained by a dual-flow heater (Supertech Instruments). The pipette solution contained (in mM): 110 K-gluconate, 4 NaCl, 20 HEPES, 0.1 EGTA, 10 phosphocreatine, 2 ATP, 0.3 GTP, 3 mg/ml biocytin adjusted to pH 7.3 to 7.35 using KOH (285 to 295 mOsm/L). Pipette resistances were 3 to 6 MΩ when filled with pipette solution. Visualization of slices and selection of cells was done under an upright microscope (BX61WI; Olympus, Tokyo, Japan equipped with infrared-differential interference contrast optics and a UV lamp). FluoSpheres (HS cells)- or eYFP (OLM cells)-labeled cells from CA1 str. oriens were selected. Only cells located deeper than approximately 50 μm measured from

the slice surface were targeted. Recordings were performed with a Multiclamp 700B amplifier (Molecular Devices). All cells were initially in voltage-clamp mode and held at −65 mV holding potential during the formation of the gigaseal. Series resistance was constantly monitored after the whole-cell configuration was established, and individual recordings taken for analysis showed stability in series resistance between a 20% margin during the whole recording.

To characterize electrophysiological properties of the cells, responses to a pulse-train of current steps was recorded in current-clamp mode, while the cells membrane potential was held around −65 mV at baseline (current values of steps in pA: 20, −20, 40, −40, 60, −60, 80, −80, 100, −100, 120, −120, 140, −140, 160, −160, 180, −180, 200, 220, 240, 260, 280 with 1,000 ms in between and 300, 400, 500, and 600 with 5,000 ms in between, each stimulus lasted for 800 ms). The recorded cells were filled with biocytin. After the recording, the slices were fixed in 4% paraformaldehyde in PB for at least 12 h, followed by washout with PB several times.

### Digital signal processing, analysis, and statistics for in vitro experiments

Data were digitized at 20 kHz with a DAQ board (National Instruments, USB-6353) and recorded with a custom software developed in C#.NET and VB.NET in the laboratory. All data were recorded, processed, and analyzed offline using standard built-in functions of Python (2.7.0.), Matlab, or in custom made software developed in Delphi.

### Fluorescent immunohistochemistry and microscopy

Hippocampal slices (300 μm thick) and sections (60 μm thick) were washed in 0.1 M PB (pH 7.4) and incubated in 30% sucrose overnight for cryoprotection. Then, they were freeze-thawed over liquid nitrogen 3 times for antigen retrieval. Slices or sections were subsequently washed in PB and Tris-buffered saline (TBS, pH 7.4), blocked in 1% human serum albumin in TBS (HSA; Sigma-Aldrich) containing 0.05% Triton X-100 (Bio-Rad Laboratories) and then incubated in a mixture of primary antibodies and reagents dissolved in TBS containing 0.05% Na-azide (Sigma-Aldrich) for 48 to 72 h (chicken anti-eGFP (1:2,000, Thermo Fisher Scientific, A10262) and/or Alexa 594-conjugated streptavidin (1:500, Molecular Probes, Cat. No. S11227) and guinea pig anti-SOM (1:500, Synaptic Systems, Cat. No. 366004). This was followed by extensive washes in TBS and incubation in the mixture of appropriate secondary antibodies overnight (Alexa 488-conjugated donkey anti-chicken (1:1,000, Jackson ImmunoResearch, Cat. No. 703-545-155), Alexa 647-conjugated donkey anti-guinea pig (1:1,000, Jackson ImmunoResearch, Cat. No. 706-605-148), and Alexa 594-conjugated streptavidin. We used DAPI staining (Sigma-Aldrich) to visualize cell nuclei. Then, slices or sections were washed in TBS and PB, dried on slides, and covered with Aquamount (BDH Chemicals) or with Vectashield Antifade Mounting Medium (Vector Laboratories). Injection sites in the MS area or hippocampus were evaluated using a Zeiss Axioplan2 microscope or a Pannoramic MIDI II slide scanner (3DHistech). Slices or sections were investigated using a Nikon A1R confocal laser-scanning microscope system built on a Ti-E inverted microscope with a 20× air objective operated by NIS-Elements AR 4.3 software. Regions of interest were reconstructed in z-stacks; the distance between the focal planes was 1.5 to 2.5 μm.

### Single immunoperoxidase and double immunogold-immunoperoxidase labeling

Sections were rinsed in PB, cryoprotected in 30% sucrose in PB overnight, and frozen in liquid nitrogen 3 times. After extensive washes in PB, sections were treated with 1% sodium borohydride in PB for 5 or 10 min (10 min in cases when both the perfusing and the immersion fixative contained glutaraldehyde). Then, sections were washed in PB and 0.05 M TBS (pH 7.4)

and blocked in 1% HSA (Sigma-Aldrich) in TBS. Then, sections of virus-injected mice were incubated in a solution of chicken anti-eGFP (1:2,000, Thermo Fisher Scientific, A10262) and mouse anti-gephyrin (1:100, Synaptic Systems, 147021) primary antibodies diluted in TBS containing 0.05% sodium azide for 2 days. After repeated washes in TBS, the sections were incubated in a blocking solution (Gel-BS) containing 0.2% cold water fish skin gelatin and 0.5% HSA in TBS for 1 h. Next, sections were incubated in a mixture of Nanogold-Fab' goat anti-mouse IgG (1:100, Nanoprobes, 2002) and goat anti-chicken biotinylated secondary antibodies (1:200, Vector Laboratories, BA-9010) in Gel-BS overnight at 4˚C.

Sections of BDA-injected mice prepared for electron microscopic analysis were treated similarly, except that anti-eGFP primary and biotinylated secondary antibodies were not added to the solutions. After extensive washes in TBS, the sections were treated with 2% glutaraldehyde in 0.1 M PB for 15 min to fix the gold particles into the tissue. After TBS washes, sections were incubated in Elite ABC (1:300, in TBS) overnight or for 2 days. To enlarge immunogold particles, sections were incubated in silver enhancement solution (SE-EM; Aurion) for 30 to 40 min at room temperature.

In the case of mice used for Neurolucida drawing of whole dendritic trees ($n = 6$), sections were treated similarly except that they were not incubated in anti-gephyrin primary- and gold-conjugated secondary antibodies, GEL-BS-, glutaraldehyde-, and SE-EM solutions. The immunoperoxidase reaction was developed using 3,3-diaminobenzidine (DAB; Sigma-Aldrich) (all virus-injected mice and $n = 5$ BDA-injected mice) or ammonium nickel sulphate-intensified DAB ($n = 1$ BDA-injected mouse used for Neurolucida drawing of whole dendritic trees) as chromogen (Fig 1).

## Specificity of antibodies

Antibodies used in this study were extensively tested for specificity. The anti-eGFP antibody did not give labeling in animals that were not injected with eGFP-expressing viruses. The anti-gephyrin antibody is KO-verified (manufacturer's information) and it systematically labeled only synapses that are typical Type II synapses. The anti-SOM antibody preferentially recognizes somatostatin-28 with minor cross-reactivity to the unprocessed precursors and does not bind to somatostatin-14 (manufacturer's information). In addition, anti-SOM antibody labeled only a subpopulation of interneurons in str. oriens, further confirming its specificity. The secondary antibodies were extensively tested for possible cross-reactivity with the other secondary or primary antibodies, and possible tissue labeling without primary antibodies was also tested to exclude auto-fluorescence or specific background labeling by the secondary antibodies. No specific-like staining was observed under these control conditions.

## Sample preparation for electron microscopy

The sections were dehydrated and contrasted with 2 different methods optimized for getting a high contrast for SEM or for more complete visualization of cells in the light microscope. For electron microscopic reconstructions of dendritic segments and somata, a modified protocol of Deerinck and colleagues was performed [108]. After washes in PB, sections were postfixed in 1% osmium-tetroxide reduced with 0.75% potassium ferrocyanide in PB on ice for 30 min and at room temperature for another 30 min. After extensive washes in distilled water (DW, $5 \times 3$ min), sections were incubated in 1% aqueous uranyl-acetate in dark for 30 min. After $5 \times 3$ min DW washes, Walton's lead aspartate staining was performed at 60˚C for 30 min. The Walton's lead aspartate solution consisted of 0,066 g lead-nitrate dissolved in 10 ml of aspartic acid stock solution. The stock solution was prepared by dissolving 0.998 g L-aspartic acid in 250 ml DW, then the pH was adjusted to 5.5 with 1N KOH. After $5 \times 3$ min DW

washes, the sections were dehydrated through ascending concentration series of ethanol (30% EtOH 3 min, 50% EtOH 5 min, 70% EtOH 2 × 5 min, 90% EtOH 2 × 5 min, absolute ethanol 2 × 7 min) and then infiltrated with acetonitrile for 2 × 7 min (first on ice, second at room temperature). The sections were transferred into aluminum boats and infiltrated with embedding resin (Durcupan, ACM; Fluka) overnight. The next day, sections were mounted on glass slides in the proper orientation, covered with coverslips, and baked at 60°C for 48 h. The sections used for complete cell reconstructions were treated with 0.5% OsO4 in 0.1 M PB for 20 min at room temperature, dehydrated in ascending alcohol series and in acetonitrile, and embedded in Durcupan. During dehydration, these sections were treated with 1% uranyl acetate in 70% ethanol for 20 min. The latter protocol gives weaker contrast for SEM, but the finer dendritic and axonal processes are more visible in the light microscope, therefore, it is more optimal for Neurolucida drawing.

## Light microscopic reconstruction of dendritic arbors

The dendritic tree of a neuron consists of segments of distinct diameter which can be classified based on their branching order and distance from the soma. For determining these parameters for each dendritic segment sampled for EM measurements, dendritic arbors were reconstructed at the light microscopic level using the Neurolucida system attached to a Zeiss Axioscope2 microscope using 100× oil-immersion objective (MBF Bioscience). For electron microscopic analysis excellent preservation of ultrastructure was required therefore the fixative used for immersion contained 0.2% to 0.25% GA. In these mice, however, some of the very distal dendrites of labeled cells could not be followed until their natural ends. Therefore, for Neurolucida reconstruction of complete dendritic trees, another group of mice were immersion-fixed using a mild fixative containing 4% PFA, but no GA. Axon arbors of HS cells could be sufficiently labeled and reconstructed only in this later group of mice. BDA injection into medial septal area resulted in retrogradely labeled HS cells. In the virus-injected Chrna2-Cre mice, we selected cells from CA1 stratum (str.) oriens and alveus with horizontally oriented dendrites that were reported to be exclusively OLM cells in this Cre strain [33]. Heavily labeled cells were selected from the CA1 area and reconstructed through serial sections ($n$ = 5 HS-, $n$ = 4 OLM-cells for electron microscopic sampling of dendrites from $n$ = 4–4 mice, and $n$ = 4 HS-, and $n$ = 4 OLM cells for complete dendritic tree reconstruction from $n$ = 2–2 mice). The distance between the soma and specific reference points of the selected dendritic segments (a branching point or a point where the dendrite leaves a given section) and the order of the segments were determined using the Neurolucida Explorer software (MBF Bioscience).

## Block preparation and serial sectioning

After the light microscopic reconstruction of the HS- and OLM cells, dendritic segments were selected at various distances from the soma and serial light microscopic z-stack images were taken in order to find them in the electron microscope. The tissue containing the selected dendritic segments was cut out using a scalpel and mounted on the top of resin blocks with cyanoacrylate glue. Ultrathin (70 nm thick) sections were cut with a diamond knife (Diatome, Biel, Switzerland) mounted on an ultramicrotome (EM UC6, Leica, Wetzlar, Germany). Ribbons of consecutive ultrathin sections ($n$ = 100–500 sections/sample) were transferred to silicon wafers (Ted Pella). Partial soma membranes of labeled cells and their afferent synapses ($n$ = 4 OLM and $n$ = 4 HS cells) were reconstructed similarly to the dendritic segments. For whole soma volume measurements serial 250 nm thick sections were prepared ($n$ = 3 OLM and 3 HS cells). Selected axonal segments of the reconstructed HS cells were analyzed with correlated light and electron microscopy similar to the

dendritic segments. Due to the larger number of labeled OLM cells than that of HS cells, their axons gave much denser labeling. In addition, OLM cells were labeled locally. Therefore, for electron microscopic investigation of OLM axons, we used random sampling instead of reconstructing individual axons. Blocks of str. lacunosum-moleculare of the injected area of CA1 were re-embedded and serially sectioned at 70 nm thickness. Randomly selected areas were imaged through serial sections ($n$ = 100–252 sections) and labeled axonal segments found in these series were reconstructed. The sections were imaged with SEM (FEI Apreo, Eindhoven, the Netherlands) in backscattered mode.

### SEM imaging

We used an FEI Apreo field emission gun SEM (Thermo Scientific) and a T1 in column detector to record the backscattered electrons. The micrographs were acquired with array tomography plugin in MAPs software (version: 3.10), which can facilitate the automatic acquisition of SEM images. The following imaging parameters were used: 1- or 4-mm working distance at high vacuum with 0,2 nA beam current, 2,5 kV high voltage, 3,5 μs/pixel dwell time, and 3 or 6 nm pixel size. The micrographs were 6,000 × 4,000 or 4,000 × 3,000 pixels and we recorded approximately 2 to 300 sections/dendritic segment with 70 nm thickness. Synapses were analyzed only from the stacks imaged using 3 nm pixel size.

### Correction factor 1, for tissue volume changes: Perfusion shrinkage

For the electron microscopic (EM) examinations, we had to preserve the ultrastructure of the brain tissue by chemical fixation. However, fixation caused shrinkage of the brain tissue; therefore, we calculated the degree of the shrinkage with the following procedure. Firstly, after terminal anesthesia, but without perfusion, we measured the volume of the brain (severed at a standard pontin coordinate) of 6 C57Bl/6NTac WT mice. After the chemical fixation, we took out the other mouse brains (severed at the same standard pontin coordinate), we washed out the fixative solution with PB several times and we compared the volume of the fixed brains and the volume of the unfixed WT brains with a graduated cylinder. We used the ratio of the volume of the native and fixed brain as a fixation shrinkage correction factor for all brains.

### Correction factor 2, for tissue volume changes: Immunohistochemistry and dehydration

Firstly, after Vibratome sectioning of the fixed brain tissue, the sections were cut into trapezoid shapes, mounted in PB without coverslip, and were imaged using a Zeiss Axioplan2 microscope in bright field mode. This was the reference size. Following immunolabeling, dehydration, and embedding in epoxy resin, these trapezoid sections were imaged again. We measured the areas of the same sections before and after processing by Fiji software [109]. The ratio between the reference and the processed section area was used to compensate for changes due to tissue processing.

### Correction factor 3, for EM section compression during diamond knife sectioning

The force applied by the diamond knife during the cutting of serial EM sections compressed the sections. Section compression was calculated by dividing the width of the tissue block face in the resin with the width of EM sections measured perpendicular to the diamond knife blade (X–Y plane). The measurements were performed by Zeiss Axioplan2 light microscope. There

was no change in the EM section dimension in the direction parallel (X plane) with the diamond knife edge.

## Applying correction factors 1–3

All dendritic segments were reconstructed from the series of SEM images. The calibrated 3D reconstructed software models of the dendritic segments and their synapses were corrected in one direction perpendicular to the edge of the diamond knife based on correction factor 3. Then, the full volumes of the models were corrected using correction factors 1 and 2. This procedure resulted in calibrated 3D models that were corrected for all foreseeable changes in tissue volume and likely produced dendrites with real-life dimensions.

## Image analysis

After SEM image series were collected, image post-processing was done in Fiji ImageJ [109]. SEM micrographs were inverted and corrected using Gaussian blur and unsharp mask to reduce noise. The SEM stacks were imported into Fiji TrakEM2 plugin15 and finely aligned. The dendrites and mitochondria were segmented using the IMOD (version 4.9) package [110]. Contouring was performed on a Wacom Cintiq 27QHD Creative Pen and display tablet. After segmentation, a 2.1*4.2*6.3 micrometer cuboid was inserted into every model as a 3D scale bar and every object was meshed with a cap in the model view of IMOD. Then, the models were exported in.obj format with the command line program imod2obje.exe and imported into Blender (version 2.79b). First, the reconstructed dendritic segments were cut into pieces of dendrites based on the order of the dendritic segments. Then, each dendrite segment was exported into *vtp* format with the NeuroMorph addon of Blender and was imported into Orobix VMTKlab, where a centerline was created for the segment. The centerline was imported back into Blender to calculate several cross-sectional areas of the dendrites. The results were exported from the NeuroMorph addon to an Excel table for further analysis. The cytoplasmic volume was calculated from whole cell reconstructions by subtracting the volume of mitochondria and nuclei from the somatic volume. Mitochondrial volume was estimated by reconstructing all mitochondria in representative sections systematically.

## Identifying dendritic orders

Dendritic segments were selected randomly from different types of dendrites. We selected dendritic segments from different distances from the somata, so that at least every 100 μm step from the somata is represented by at least a few dendritic segments. We also tried to sample from dendrites with all the different orders. The order of the dendritic segment was defined based on the following rule: first-order dendrites originated from somata. After first-order dendrites bifurcate, both offspring dendrites became second-order dendrites. All third, fourth . . . nth order dendrites only originated from second, third, and (n-1)th order dendrites, respectively, and so on.

## Technical validation of 3D measurements

To validate our 3D measurements, we created artificial dendrite models and shapes, the volumes and surfaces of which were known. Then, using Blender, we artificially sectioned them at 70 nm and rendered them (equivalent to real-life photography). Each rendered image section was set to the same fixed pixel resolution that we used for real SEM photos; thus, we made EM-like image stacks, which were segmented using iMOD. After segmentation, we meshed

the objects as close type, exported them as *obj* files, and processed and measured them in Blender. Finally, we compared the original and measured volumes and surface areas, and the differences were negligible. With regards to the final measurements, dendritic model segments were analyzed using both Blender software directly and a NeuroMorph software plugin (both are open access) and the results were very similar. After the validation of the closed model's volume and surface, we also made an artificial model of synapses. We carried out the same procedure as above, but the synapses were meshed as open-type objects. We found that we get the statistically most precise result, if we draw the last layer of the synapse in the next layer as a double-layered cap. This resulted in a highly precise and unbiased estimation of the small synaptic areas.

## Extraction of electrophysiological features

To extract electrophysiological features from our current-clamp recordings, we used a modified version of the BluepyEfe software tool (https://github.com/BlueBrain/BluePyEfe). Our modification involved the definition and implementation of standardized features: instead of comparing the responses of the cells to current steps of specific amplitudes, we analyzed the responses to current steps where similar behaviors were expected. The following current steps were used: (i) rheobase current step: the smallest current step where the cell fired at least 1 action potential; (ii) standard negative current step: usually $-0.1$ nA, or the closest one in the protocol; (iii) steady state: the smallest current step where the cell fires a minimum of 8 action potentials; and (iv) maximal activity: where the cell fired the most action potentials.

We performed feature extraction for every measurement, even when more than one measurement came from the same cell. All features that are available in the Electrophys Feature Extraction Library (eFEL) package (https://github.com/BlueBrain/eFEL, versions 4.0.4 and 4.1.49) were extracted, and later those that gave misleading or meaningless results were filtered out and were not considered during further evaluation. In the statistical analysis, a total of 267 features were used.

The statistical analysis of the electrophysiological characteristics was performed with Python. The Dataframe object of the Pandas library (Pandas Development Team, 2020) was used to store and process data and Matplotlib was used for generating figures [111].

## Modeling: Construction of multi-compartmental models

The 4 OLM and 4 HS cell morphologies were fully reconstructed in Neurolucida with realistic dendritic diameters. The morphologies were then further processed using the Neuron simulator [112] and were spatially discretized using the d-lambda rule. It resulted in compartments that are called "segments" later on and have electronic length which was less than 0.1 times the space constant (lambda). This was calculated in a neuron-, location-, and state-specific manner.

Two types of functional models were constructed using these structural models. The simpler, essentially passive models were designed to capture subthreshold behavior. They contained a leak conductance and a hyperpolarization-activated conductance (Ih), which were distributed uniformly in all soma-dendritic compartments. The model of Ih was based on [63], using both experimentally measured kinetic components.

The active models were designed to capture both the subthreshold and the suprathreshold (spiking) behavior of OLM and HS cells. In these models, the properties of voltage-gated conductances and their distributions were based on a model of OLM cells described by Lawrence and colleagues [68]. This model contains 8 active conductances: sodium channels, fast and

slow delayed-rectifier potassium channels, M-type potassium channels, A-type potassium channels, calcium-dependent potassium channels, T-type calcium channels, and Ih. The model also contains 2 types of leak current, whose conductances are set separately for the soma and axon. The sodium channels also had different densities in the somatic, axonal, and dendritic compartments. We fitted these 14 parameters so that the model behavior showed the best possible match to our experimental data (see below). Passive parameters of the model were fixed to the values given by the original model of Lawrence and colleagues (the specific membrane capacitance was 1 $\mu$F/cm$^2$ and the axial resistance was 300 Ohm cm). Simulations were carried out by the Neuron simulator. All models are available in the ModelDB database (https://modeldb.science).

## Modeling: Parameter tuning

The Neuroptimus software [113,114] is an interactive Python tool for the automated fitting of unknown parameters between given boundaries. From the several algorithms implemented, we used the Classical Evolutionary Strategy (CES) from the Inspyred package along with feature-based error function based on the eFEL feature extraction library (https://github.com/BlueBrain/eFEL). We used a population size of 128 and ran the algorithm for 100 generations. We used supercomputers available through the Neuroscience Gateway [115] to perform the optimization.

For each morphology, we performed 20 parallel parameter searches in the case of the passive models and 10 parallel parameter searches in the case of the active models between realistically constrained boundaries, and then chose the parameter sets with the smallest fitness values. The chosen parameter sets were representative of the whole population.

## Modeling: Model testing

HippoUnit [116] is a Python module that automatically performs simulations that mimic experimental protocols and compares the results to experimental data. It is based on NeuronUnit and SciUnit [117]. It uses models built in NEURON as a user-defined Python class. It has 5 built-in tests, including the Somatic Feature Test, which uses the spiking features of eFEL (https://github.com/BlueBrain/eFEL), the Back-propagating Action Potential Test, the Post-synaptic Potential Attenuation Test, the Depolarization Block Test, and the Oblique integration Test.

For this study, we used the Somatic Feature Test to validate our fitted models against the experimental data, and a modified version of the Post-synaptic Potential Attenuation Test was used to evaluate the signal propagation properties of the models. The configuration file was matched to the experimental data that defines the size and shape of one EPSC [33,64]. A modified version of the test was also created to use IPSC data [13] instead of excitation.

## Modeling: Synaptic inputs

To define the number of synapses on each segment of the models, the electron microscopic data was analyzed by MATLAB software [118]. Multiple linear regression was performed with different combinations of the measured properties of dendritic segments as predictors, and the number of synapses per membrane area was best predicted by combining the dendritic segment's distance from the soma and the diameter of the segments. We calculated the predicted synapse number for each segment of the reconstructed morphologies with the following equation:

$$(a + b * distance + c * diameter) * membrane\ area\ of\ the\ segment, \tag{1}$$

where a, b, and c are coefficients of the multiple linear regression that are different in each specific case (the number of excitatory and inhibitory synapses on HS versus OLM cells).

The equation above, without the multiplication by area, defines the density of synapses, which was also used in the calculation of the time-averaged membrane conductance density created by synaptic inputs, according to the equation:

$$\text{Conductance per area} = \text{synapse number per area} * \text{rate} * \text{peak conductance} * \text{tau},$$

where rate is the firing rate of the presynaptic neurons, and peak conductance and tau characterize the individual synaptic input events (see Results for further details). In our simulations of the neurons in the presence of background synaptic input (which we refer to as the high-conductance state of the neurons), we used these equations to calculate the contributions of excitatory and inhibitory synapses to the total membrane conductance.

These time-averaged excitatory and inhibitory conductances were implemented as the leak conductance in separate NMODL files, and their exact amount was stated explicitly for each segment in the models in the case of the passive models.

In the passive and active models, when single synaptic inputs were modeled, we used Neuron's built in Exp2Syn object.

To evaluate how the spiking activity of the model neurons depends on the amount of synaptic input they receive, we stimulated the cells with 20, 40, 60, 80, 100, 200, 300, 400, and 500 synapses activated using independent Poisson spike trains at 10 Hz at random dendritic locations and measured the spiking rate. First the locations of the synapses were selected from the first 100 μm from the soma, then from 300 to 400 μm from the soma, and lastly from the 0 to 400 μm range that represented almost the whole dendritic arbor. Then, we divided the synapse number with the total number of excitatory synapses in the given dendritic area of each cell and multiplied by 100 to get the percentage of the stimulated synapses from the total possible excitation. To examine the response to theta frequency stimulation, we stimulated 5% to 15% of the excitatory synapses randomly distributed on the whole dendritic arbor for 30 s, using an input rate that varied between 1 and 21 Hz in a sinusoidal manner at the 7 Hz theta frequency.

## Statistical considerations

Because we have generated several large data sets and made several comparisons, we had to consider the statistical problem of multiple comparisons with respect to a type I error. However, in the case of morphological statistics, no correction was made because these comparisons were already decided a priori and because all morphological statistical results are reported regardless of their significance.

However, for the simultaneous comparison of 267 in vitro physiological features of the 2 cell types, we report the results only for the significantly different features (in addition to a few preselected basic ones), and therefore, we used a simple but rigorous Bonferroni correction procedure (we divided $p = 0.05$ by 267). When $p$-values reached the significance level even after correction, they were highlighted in red, but when they were significant only without correction, we highlighted their values in blue in Table 4 and S11 Data.

## Supporting information

**S1 Fig. Local axon arbors of HS cells.** Neurolucida reconstructions of BDA-labeled HS cells. Complete dendritic trees are shown in green, whereas partially reconstructed local axons are shown in black. Inset shows the location of cells in str. oriens of the hippocampal CA1 area. The axonal segments indicated with green highlighting and dashed circles were reconstructed

and analyzed in 3D using a scanning electron microscope.
(JPG)

**S2 Fig. Proportions of OLM and HS cells in somatostatin-positive interneurons of CA1 str. oriens.** A1-3: Confocal laser-scanning microscope image of a Cre-dependent AAV-eYFP tracer injection site in the CA1 str. oriens of a Chrna2-Cre mouse. The section was double-labeled for somatostatin and eYFP, and 49.1%, 42%, and 37.9% of somatostatin-positive cells were eYFP-positive OLM cells in 3 mice, respectively. B1-3: Confocal laser-scanning micro-scope image from CA1 str. oriens in a mouse with FluoSpheres-injection into the medial sep-tum. Yellow-green fluorescent microbeads were retrogradely transported into the somata of HS cells, whereas somatostatin was labeled by immunocytochemistry; 9.5%, 8.1%, and 7.5% of somatostatin-positive cells were tracer-containing HS cells in 3 mice, respectively. Yellow arrowheads indicate double-labeled cells, whereas red arrowheads show somatostatin-positive cells that did not contain tracer. Scale bar: 50 μm.
(JPG)

**S3 Fig. Sholl analysis of OLM and HS cell dendritic trees.** (A) Dendritic trees of an OLM (magenta) and an HS cell (green) with overlaid concentric spheres. The radius increment of each sphere is 50 μm. (B–H) Morphological parameters of OLM (magenta) and HS cells (green) mea-sured 3 dimensions in concentric spherical shells 50–550 μm from soma. Data points represent single cells. (B) Number of dendrites intersecting a sphere. (C) Total dendritic length in shells. (D) Total surface area of dendrites in shells. (E) Total volume of dendrites in shells. (F) Average diam-eter of dendrites in shells. (G) Number of branching points (nodes) in shells. (H) Number of den-drites ending in a shell. The analysis did not reveal differences, except that HS cells had branches in shells further than 500 μm, while the dendrites of the OLM cells have not reached that far from their somata. See raw data (exported directly from Neurolucida Explorer) in S20 Data.
(JPG)

**S4 Fig. Distribution of the best-fitting parameters for each cell morphology.** The median and interquartile values (q1: 25%, q2: 75%) of the best-fitting parameters of the 20 optimiza-tion runs for each cell morphology are indicated by the boxes. Whiskers extend to the smallest and largest value within the kernel of q1-1.5* interquartile range and q2+1.5* interquartile range. Those values that are not in that kernel are defined as outliers and represented as circles. See raw data in S24 Data.
(JPG)

**S5 Fig. Comparison of the values of the electrophysiological features extracted from HS cell recordings and from the overall best model of each HS morphology.** The mean of the experimental data is marked with black X with its corresponding standard deviation. Each cell is represented by its associated color. Only the features corresponding to hyperpolarizing cur-rent injections were used in this validation. In the Electrophys Feature Extraction Library (eFEL), the exact feature names are the following: Steady state voltage after stimulus end: stea-dy_state_voltage; Steady state voltage before stimulus end: steady_state_voltage_stimend; Voltage base: voltage_base; Voltage deflection: voltage_deflection; Voltage deflection (Lib.: vs_sse): voltage_deflection_vbsse; Voltage deflection begin (at the beginning of the stimulus): voltage_deflection_begin; Sag ratio: sag_ratio1; Sag amplitude: sag_amplitude; Sag time con-stant: sag_time_constant. The data for this figure is available in the folder Codes_and_scripts \hippounit\somatic_feature_results\Simplified_models\Chosen_HS_somafeatures_optimized at https://zenodo.org/records/10580838.
(JPG)

**S6 Fig. Comparison of the values of the electrophysiological features extracted from OLM cell recordings and from the overall best model of each OLM morphology.** The mean of the experimental data is marked with black X with its corresponding standard deviation. Each cell is represented by their associated color. Only the features corresponding to hyperpolarizing current injections were used in this validation. In the Electrophys Feature Extraction Library (eFEL), the exact feature names are the following: Steady state voltage after stimulus end: steady_state_voltage; Steady state voltage before stimulus end: steady_state_voltage_stimend; Voltage base: voltage_base; Voltage deflection: voltage_deflection; Voltage deflection (Lib.: vs_sse): voltage_deflection_vbsse; Voltage deflection begin (at the beginning of the stimulus): voltage_deflection_begin; Sag ratio: sag_ratio1; Sag amplitude: sag_amplitude; Sag time constant: sag_time_constant. The data for this figure is available in the folder Codes_and_scripts \hippounit\somatic_feature_results\Simplified_models\Chosen_OLM_somafeatures_optimized at https://zenodo.org/records/10580838.
(JPG)

**S7 Fig. Local reversal potential of dendritic segments in the high-conductance state of passive models.** The local reversal potential of dendritic segments as a function of their distance from the soma, calculated from the estimated average excitatory, inhibitory, and leak currents in the high-conductance state of each passive model neuron.
(JPG)

**S8 Fig. Comparison of the values of the electrophysiological features extracted from OLM cell recordings and from the overall best model of each OLM morphology.** The mean of the experimental data is marked with black X with its corresponding standard deviation. Each cell is represented by its associated color. Sixteen features were selected for this figure to represent both spiking and non-spiking behavior of the cell without redundancy from the HippoUnit's Somatic Features Test. The data for this figure is available in the folder Codes_and_scripts\hippounit\somatic_feature_results\Detailed_models\Validation_chosen_ones_olm at https://zenodo.org/records/10580838.
(JPG)

**S9 Fig. Comparison of the values of the electrophysiological features extracted from HS cell recordings and from the overall best model of each HS morphology.** The mean of the experimental data is marked with black X with its corresponding standard deviation. Each cell is represented by its associated color. Sixteen features were selected for this figure to represent both spiking and non-spiking behavior of the cell without redundancy from the HippoUnit's Somatic Features Test. The data for this figure is available in the folder Codes_and_scripts\hippounit\somatic_feature_results\Detailed_models\Validation_chosen_ones_hs at https://zenodo.org/records/10580838.
(JPG)

**S10 Fig. Spiking response of active neuronal models to scattered synaptic input.** (A–C) The spiking rates of model neurons as a function of the number of activated synaptic inputs. (D–F) The same data shown as a function of the percentage of activated synapses in the given cell and distance range. Synapses were selected randomly from different distance ranges from the soma: (A, D) 0–100 μm; (B, E) 300–400 μm; (C, F) 0–400 μm.
(JPG)

**S11 Fig. Modulation of the spiking rate of model neurons by theta-periodic synaptic input.** Spike rates of active model cells as a function of the phase (in degrees) of the 7 Hz theta oscillation that modulated the rate of synaptic inputs; 5% of the total number of excitatory synapses

were randomly distributed in the whole dendritic arbor in all cells except OLM_1811, where the percent of activated synapses was increased to 15% to ensure sufficient activity. See raw data in S25 Data.
(JPG)

**S1 Data. IDs of dendritic segments in Fig 2 and S2–S5 Data.**
(XLSX)

**S2 Data. Raw data of OLM dendritic segments.**
(XLSX)

**S3 Data. Raw data of HS dendritic segments.**
(XLSX)

**S4 Data. Raw data of OLM branching points.**
(XLSX)

**S5 Data. Raw data of HS branching points.**
(XLSX)

**S6 Data. Raw data of somata.**
(XLSX)

**S7 Data. Raw data of axons.**
(XLSX)

**S8 Data. Comparisons of anatomical properties between OLM and HS cells.**
(XLSX)

**S9 Data. Comparison of anatomical properties within cell types.**
(XLSX)

**S10 Data. Correlations between anatomical properties within cell types.**
(XLSX)

**S11 Data. Physiological and model parameters and statistics.**
(XLSX)

**S12 Data. Coefficients used to calculate the No. of syn. of each segment of the model.**
(XLSX)

**S13 Data. The best-fitting parameters for each morphology.**
(XLSX)

**S14 Data. Synapse parameters used in the models to calculate synaptic conductances.**
(XLSX)

**S15 Data. Neurolucida Explorer dendrite segment analysis.**
(XLSX)

**S16 Data. Active parameters.**
(XLSX)

**S17 Data. Calculated diameters of OLM dendritic segments.**
(XLSX)

**S18 Data. Calculated diameters of HS dendritic segments.**
(XLSX)

**S19 Data. Identifications of spine/shaft synapses.**
(XLSX)

**S20 Data. Sholl analysis of OLM and HS cell dendritic trees.**
(XLSX)

**S21 Data. Reconstructions of dendritic trees directly exported from Neurolucida.**
(ZIP)

**S22 Data. Dendrograms of OLM and HS cell dendritic trees.**
(ZIP)

**S23 Data. Calculated number of excitatory and inhibitory synapses on each dendritic segments.**
(XLSX)

**S24 Data. Fitted parameters of the simplified models.**
(XLSX)

**S25 Data. Spike times with their corresponding angles in response to theta excitation in the models.**
(XLSX)

## Acknowledgments

We thank E. Szépné Simon, N. Kriczky, K. H. Kása, and Z. Hajós for helping with the experiments; S. Sáray, M. Blazsek, and L. Tordai for contributions to data analysis and modeling; and G. Goda, K. Iványi, and A. Kriczky, for other assistance. We thank Christopher J. Kohl for the Blender script that measured the perimeters of the dendritic cross-sections. We thank the Electron Microscopy Center at the Institute of Experimental Medicine (IEM) for technical help with the FEI Apreo scanning electron microscope and the Hitachi H-7100 transmission electron microscope and S. Kőszegi for help in image acquisition. We acknowledge Light Microscopy Center at the IEM for the use of the 3DHistech Pannoramic MIDI II slide scanner and the Nikon A1R confocal laser-scanning microscope and thank P. Vági and L. Barna for their assistance in image acquisition. We thank Z. Erdélyi, F. Erdélyi, and the staff of the Animal Facility and the Medical Gene Technology Unit of IEM for expert technical help with the breeding and genotyping of the mouse strains used in this study. We would like to thank M. Sümegi and E. Sipos, the Virus Technology Unit of IEM, for technical support.

## Author Contributions

**Conceptualization:** Virág Takács, Zsuzsanna Bardóczi, Áron Orosz, Abel Major, Luca Tar, Péter Berki, Szabolcs Káli, Gábor Nyiri.

**Data curation:** Virág Takács, Zsuzsanna Bardóczi, Áron Orosz, Abel Major, Luca Tar, Péter Berki, Szabolcs Káli.

**Formal analysis:** Virág Takács, Zsuzsanna Bardóczi, Áron Orosz, Abel Major, Luca Tar, Péter Berki, Péter Papp, Márton I. Mayer, Hunor Sebők, Luca Zsolt, Katalin E. Sos, Szabolcs Káli, Gábor Nyiri.

**Funding acquisition:** Szabolcs Káli, Tamás F. Freund, Gábor Nyiri.

**Investigation:** Virág Takács, Zsuzsanna Bardóczi, Áron Orosz, Abel Major, Luca Tar, Péter Berki, Péter Papp, Márton I. Mayer, Hunor Sebők, Katalin E. Sos.

**Methodology:** Virág Takács, Zsuzsanna Bardóczi, Áron Orosz, Abel Major, Luca Tar, Péter Berki, Szabolcs Káli, Gábor Nyiri.

**Project administration:** Virág Takács, Szabolcs Káli, Gábor Nyiri.

**Supervision:** Szabolcs Káli, Tamás F. Freund, Gábor Nyiri.

**Validation:** Zsuzsanna Bardóczi, Áron Orosz, Abel Major, Luca Tar, Péter Berki, Szabolcs Káli.

**Visualization:** Virág Takács, Zsuzsanna Bardóczi, Áron Orosz, Abel Major, Luca Tar, Péter Berki, Szabolcs Káli, Gábor Nyiri.

**Writing – original draft:** Virág Takács, Zsuzsanna Bardóczi, Áron Orosz, Abel Major, Luca Tar, Péter Berki, Hunor Sebők, Szabolcs Káli, Gábor Nyiri.

**Writing – review & editing:** Virág Takács, Zsuzsanna Bardóczi, Áron Orosz, Abel Major, Luca Tar, Péter Berki, Péter Papp, Márton I. Mayer, Hunor Sebők, Luca Zsolt, Katalin E. Sos, Szabolcs Káli, Tamás F. Freund, Gábor Nyiri.

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
