## [Editor Report · Decision Letter 0]

24 Jul 2023

Dear Dr Nyiri, 

Thank you for submitting your manuscript entitled "Synaptic and dendritic architecture of two types of hippocampal somatostatin interneurons" for consideration as a Methods and Resources by PLOS Biology.

Your manuscript has now been evaluated by the PLOS Biology editorial staff as well as by an academic editor with relevant expertise and I am writing to let you know that we would like to send your submission out for external peer review.

Once your full submission is complete, your paper will undergo a series of checks in preparation for peer review. After your manuscript has passed the checks it will be sent out for review. To provide the metadata for your submission, please Login to Editorial Manager (https://www.editorialmanager.com/pbiology) within two working days, i.e. by Jul 26 2023 11:59PM.

Kind regards,

Christian

Christian Schnell, PhD

Senior Editor

PLOS Biology

cschnell@plos.org

---

## [Decision Letter · Decision Letter 1]

3 Sep 2023

Dear Dr Nyiri,

Thank you for your patience while your manuscript "Synaptic and dendritic architecture of two types of hippocampal somatostatin interneurons" was peer-reviewed at PLOS Biology. It has now been evaluated by the PLOS Biology editors, an Academic Editor with relevant expertise, and by several independent reviewers. 

In light of the reviews, which you will find at the end of this email, we would like to invite you to revise the work to thoroughly address the reviewers' reports.

As you will see below, the reviewers that the reviewers think that the study is very well executed and provides important insights. However, R1 and R2 in particular raise some concerns that will require additional experimental/computational work to address.

Given the extent of revision needed, we cannot make a decision about publication until we have seen the revised manuscript and your response to the reviewers' comments. Your revised manuscript is likely to be sent for further evaluation by all or a subset of the reviewers.

**IMPORTANT - SUBMITTING YOUR REVISION**

*Re-submission Checklist*

*Published Peer Review*

*PLOS Data Policy*

*Blot and Gel Data Policy*

Sincerely,

Christian

Christian Schnell, PhD

Senior Editor

PLOS Biology

cschnell@plos.org

REVIEWS:

Reviewer's Responses to Questions

PLOS authors have the option to publish the peer review history of their article (what does this mean?). If published, this will include your full peer review and any attached files.

Reviewer #1: No

Reviewer #2: No

Reviewer #3: Yes: Frances K Skinner

Reviewer #1: The work by Takács and colleagues is detailed quantitative study of the morphological and physiological characteristics, distribution and size of putative glutamatergic excitatory and GABAergic inhibitory synaptic inputs along the entire somato-dendritic domain of Hl and OLM cells in CA1. The main findings are: (1) dendrites were highly variable in thickness in both Hl than OLM cells; (2) the density and number of glutamatergic and GABAergic inputs was higher in HL than OLM cells; (3) synaptic inputs were larger at proximal than distal dendrites of HL cells but not OLM cells; in OLM neurons glutamatergic inputs increased in size with distance from the soma; (4) the size of output synapses of OLM cells was almost twice as large as those of HL cells. Finally, HL cells tended to discharge at mildly higher frequency in vitro during current injections, particularly at the beginning of the injection of a given amplitude. On the basis of anatomical reconstructions and estimates on single synaptic as well as total compartment-specific synaptic and leak conductances they further developed single cell computational models to define how the distribution of inputs may shape EPSPs and IPSPs measured at the soma in tow states: the silent and a high conductive state. As expected, EPSPs were larger in size in HL cells at silent states than OLM cells and during high conductance states, PSPs were smaller in Hl than OLM cells due to the higher total membrane conductance and stronger filtering of the signals during their propagation from the induction side to the soma. 

The manuscript is very well written and illustrated. The set of data provided in this study is very important for our understanding how neurons of similar / comparable morphology may differentially influence signal propagation. The study shows that details of input size and distribution matter. However, my enthusiasm is diminished by the lacking experimental evidences on single input conductances, which could have been measured by e.g. extracellular minimal stimulation along the somato-dendritic axis (dendritic recordings would have been ideal, however, difficult to perform). Moreover, it remains unclear how the different input properties and their distribution may contribute to spatial and temporal signal integration and the control of action potential generation in the two cell types. In other words, what is the functional difference of both cell types, how do the different input distributions and sizes manifest in potential physiological differences in signal integration? 

Main concern:

1. What is the real single input conductance in both cell types? One option would be to stimulate with minimal intensity synaptic inputs (glutamatergic / GABAergic) at distinct distances between soma and dendrite and measure IPSC / EPSC sizes. The idea behind the proposal is to get a bit more realistic estimates on the conductances used in the single cell models.

2. The here provided single cell models would be ideally suited to examine in more detail differences in the temporal and spatial synaptic integration of excitatory inputs and the influence of inhibition on the dendritic integrative processes in dependence of dendritic location. The impact of inhibition on the very distal dendrites of OLM cells should be stronger than at more proximal sides. Based on Figure 8, the E/I ratio seems to switch between proximal and distal dendrites of OLM cells but appears to be constant in HL neurons. For precise investigations of the role of inhibition in the integration of EPSPs it is of course important to have good estimates on the reversal potential of IPSCs. 

3. As stated by the authors in the discussion, more information on the impact of the synaptic input distribution on the recruitment of HL / OLM cells would be perfect. May be the authors see a possibility to make some assumptions on the distribution of voltage gated conductances similar to published data on OLM cells; although the reviewer fully understands the difficulty on making such assumptions and the difficulty to do the related experiments. 

Reviewer #2: The paper by Takács et al., titled "Synaptic and dendritic architecture of two types of hippocampal somatostatin interneurons", describes extensive profiling of two cell types present among the somatostatin (SST) positive inhibitory neurons in the hippocampus, termed OLM and HS interneurons. The authors characterize the abundance of these two neuron types among the SST interneurons. They quantify the morphologies of the two cell types, including dendritic and axonal features and synaptic targeting, for both the incoming and outgoing synapses. Further, intrinsic electrophysiology of these cells is assessed in slice recordings. Finally, the authors fit morphologically detailed models (but, as they describe them, biophysically simplified, since active conductances are excluded) for these two cell types and investigate responses of the cell models to synaptic inputs.

Overall, the paper has many appealing characteristics. It summarizes a large body of carefully obtained data. The data are multimodal, i.e., come from a variety of experimental techniques to shed light onto multiple properties of the cells. And, modeling is used to integrate the data and obtain further insights through simulations. The paper is well written and clear.

There are, however, some cons as well. To me, the major issue is that modeling did not use active conductances, therefore resulting in models that cannot generate action potentials and cannot be used for any studies beyond those using weak subthreshold stimuli. For a relatively high-profile publication, this is not sufficient, especially because the paper did not build models for that many cells (only 4 each for OLM and HS cells), and neither did it profile electrophysiology in a particularly large number of cells.

I would therefore suggest that, to make their paper sufficiently strong, the authors should add models with active conductances in the soma (and, perhaps, the axon initial segments). I understand that adding active conductances in dendrites would be way too computationally expensive, so I am not suggesting that. But, developing models with active conductances in the soma would be a necessary addition, which would improve the paper drastically.

With such an addition, the authors can simulate spiking responses to current injections into the soma as well as to synaptic inputs, for example, at theta frequencies. These simulation results will show whether OLM and HS cells exhibit the same or distinct spiking responses under such circumstances, which will be an important contribution of the modeling. As it stands now, the differences observed with modeling subthreshold phenomena are all rather minor and are likely not highly relevant for physiological situations where cells produce many action potentials.

Obviously, we don't know whether the results will be different between the OLM and HS, but it is important to explore this. That will elevate the results of the paper from less relevant to in vivo physiology to highly relevant.

Other comments.

Page 5, bottom, where synapses onto the OLM and HS cells are characterized:

It would be interesting to know how many (what fraction) of the excitatory input synapses are targeting spines vs. shaft and soma, and the same for inhibitory input synapses.

Page 9, Proportions of OLM and HS cells within the somatostatin-positive interneurons of CA1 str. oriens:

It would be good to provide numbers for each mouse, since it's only 3 in each case. One wonders whether the proportions are well conserved across animals.

Page 11, modeling subthreshold responses:

The authors only show summary statistics for modeling subthreshold responses to negative current injections in these cells. It would be useful to show in the main figures examples of the actual simulated traces, overlapped with the experimental traces.

Page 13, last paragraph:

The part about mitochondria is interesting, but more information should be shown if the authors want to include it in the paper. There should be a reference to the data on the volume occupied by mitochondria in the Tables (which are mentioned earlier, but readers would forget by now). And, where the authors say that they did not see a change in the behavior of model neurons, they should show these results in a figure or table.

Page 14, first paragraph:

These are good references, but the authors completely ignore a large body of work from the Allen Institute, which provided extensive data for constructing detailed models as well as delivered such models. The data are widely used in the community. Some of the relevant papers are as follows:

https://doi.org/10.1038/s41467-017-02718-3

https://doi.org/10.1038/s41593-019-0417-0

https://doi.org/10.1016/j.cell.2020.09.057

https://doi.org/10.1126/science.abj5861

https://doi.org/10.1016/j.neuron.2020.01.040

The same point applies to the statement and references at the very end of Discussion.

Page 19, lines 12-25:

A good starting point can be to use the same ionic conductances for the OLM cells and HS cells. As the recordings presented here show, the electrophysiological responses of these two cell types are not drastically different, suggesting that same conductances might work fine.

Fig. 1L:

"Complete Neurolucida reconstruction of OLM and HS dendritic trees and (Fig. 5, 6)."

It looks like some text is missing after the "and".

Fig. 9:

Please provide units for Density.

Table 4. "Physiological and model parameters and statistics":

These do not seem to be model parameters, or maybe I don't understand something.

Supplementary Figure 4:

It is somewhat surprising to see such different values of specific capacitance. I would expect them to be closer to 1 uF/cm^2, but that is the case only for one of the 8 cells. For the 4 HS cells, the specific capacitance tends to be small, around 0.6 uF/cm^2.

Are authors sure that these are correct/realistic values? Could they try building their models with the specific capacitance fixed at 1.0 uF/cm^2 and see if they still can reproduce the experimental electrophysiology?

In any case, it would be important to comment on such distinct values of specific capacitance and explain why this is reasonable. It is hard to see what properties of the cell membrane would make the specific capacitance so different (by a factor more than 2 between some of the cells).

Supplementary Figs. 5 and 6:

Please label the y-axis. And, it would be more intuitive to let more negative numbers to go down rather than go up.

Supplementary Figure 6:

It is surprising that the Voltage Base for OLM cells is uniformly fit worse than most other electrophysiological features. Why is that, and can authors improve the fits?

Reviewer #3: In this paper, the authors do an in depth exploration of OLM and HS somatostatin cell types, finding major differences between the density and distribution of their synaptic inputs and mitochondria. They perform 3D reconstructions using EM for dendritic segments etc. Overall, this is an expansive, detailed and careful (e.g., tissue shrinkage considerations) study that is able to uncover distinct differences between these two somatostatin cell types that can help understand their different functional contributions. However, I have some comments and suggestions below. Overall, this will be a useful resource for the community. 

GENERAL COMMENTS

1) The authors could situate their work a bit more in the expanding literature of different cell types. For example, consider the recent review of Kessaris and Denaxa (Curr Opin Neurobiol 2023) and references therein. That is, they could describe more about OLM cell types overall I thought (e.g., different types mentioned in Harris et al Plos biol 2018). They clearly describe that they identify OLM cells via Chrna2, but this is a subset as they know, and they do compute relative numbers (proportions on page 9), so important to more broadly and completely situate their work regarding these cell types.

2) Please quantify 'reasonably good'(validation on subthreshold) on page 11 - many other details are quantified and stats applied, so not sure why the authors did not provide any quantification of what they considered reasonably good - eyeballing?

3) The authors refer to 'predictive equations' on page 10 regarding synaptic distributions, but I could not find them (prediction coefficients etc.). Given the differences obtained, this would seem to be an important aspect to include fully.

4) As noted by the authors, there are large differences in PSPs when in vivo like states are used relative to in vitro. However, it is interesting that the synaptic differences between OLM and HS cells does not seem to be present anymore. Is this a fair comment? If so, I think that this should be commented on, and whether this might imply less functional differences and/or reduce what their functional differences might be? Also along functional lines, while the authors comment on spiking differences between OLM and HS cells, they do seem to have similar mean frequencies, right?

MINOR COMMENTS

- Add 'dendrites' to line 39 on page 8, as it is included on line 23.

- There are more recent OLM cell models that should be cited - Sekulic et al. Frontiers in Cellular Neuroscience 2020 (line 23, page 19).

---

## [Decision Letter · Decision Letter 2]

10 Jan 2024

Dear Gábor,

Happy New Year!

Thank you for your patience while we considered your revised manuscript "Synaptic and dendritic architecture of two types of hippocampal somatostatin interneurons" for publication as a Methods and Resources at PLOS Biology. This revised version of your manuscript has been evaluated by the PLOS Biology editors, the Academic Editor, and two of the original reviewers.

Based on the reviews and on our Academic Editor's assessment of your revision, we are likely to accept this manuscript for publication, provided you satisfactorily address the minor requests from the reviewers and following data and other policy-related requests:

* We would like to suggest a different title to improve readability: 

Synaptic and dendritic architecture of different types of hippocampal somatostatin interneurons

* Please add the links to the funding agencies in the Financial Disclosure statement in the manuscript details.

* In the ethics statement, please provide the approval number from your Institutional Animal Care and Use Committee. 

DATA POLICY:

Regardless of the method selected, please ensure that you provide the individual numerical values that underlie the summary data displayed in the following figure panels as they are essential for readers to assess your analysis and to reproduce it: 3D, 3E, 3F, 3G, 5L, 5M, 8A, 8B, 8C, 8D, S3B–S3H, S4, S5, S6, S8, S9, and S11

Please also ensure that figure legends in your manuscript include information on where the underlying data can be found, and ensure your supplemental data file/s has a legend. For example: "Source data can be found in S1_Data."

CODE POLICY

Per journal policy, as the code that you have generated is important to support the conclusions of your manuscript, we require that you make it available without restrictions upon publication. Please ensure that the code is sufficiently well documented and reusable, and that your Data Statement in the Editorial Manager submission system accurately describes where your code can be found.

If you decide to use github for code and/or data deposition, please assign a DOI so that the repository is citable and versioned for your paper. Zenodo is one of the available tools for this.

We expect to receive your revised manuscript within two weeks. 

*Published Peer Review History*

*Press*

Sincerely,

Christian

Christian Schnell, PhD

Senior Editor,

cschnell@plos.org,

PLOS Biology

Reviewer remarks:

Reviewer #1: The authors performed a substantial revision of the manuscript and addressed all points made. I am very satisfied with the revision and do not see any need for an additional revision. 

Reviewer #2: The authors did a great job addressing the reviewers' comments, which resulted in a substantially improved manuscript. It is wonderful to see the models with active conductances working quite well (though not without issues, but at least now we know what we can get with the current level of knowledge). Congratulations to the authors on such a nice body of work.

Interestingly, the features that seem to be the hardest to reproduce are related to the AHP depth. The models produce AHP that is way too deep. Perhaps this is something to think about in the future - maybe modifications to the model conductances can help.

One minor issue - please add labels to the y-axes in Supplementary Figures 8 and 9.

---

## [Editor Report · Decision Letter 3]

1 Feb 2024

Dear Gábor,

Thank you for submitting your manuscript "Synaptic and dendritic architecture of different types of hippocampal somatostatin interneurons".

I've gone through all our editorial checks and your manuscripts looks alright, except for the Methods section which seems to have gone missing during the revision. Could you please have a look and put this back in?

*Published Peer Review History*

*Press*

Sincerely,

Christian

Christian Schnell, PhD

Senior Editor

cschnell@plos.org

PLOS Biology

---

## [Editor Report · Decision Letter 4]

6 Feb 2024

Dear Gábor,

Thank you for the submission of your revised Methods and Resources "Synaptic and dendritic architecture of different types of hippocampal somatostatin interneurons" for publication in PLOS Biology. On behalf of my colleagues and the Academic Editor, Carl Petersen, I am pleased to say that we can in principle accept your manuscript for publication, provided you address any remaining formatting and reporting issues. These will be detailed in an email you should receive within 2-3 business days from our colleagues in the journal operations team; no action is required from you until then. Please note that we will not be able to formally accept your manuscript and schedule it for publication until you have completed any requested changes.

PRESS

Sincerely, 

Christian

Christian Schnell, PhD

Senior Editor

PLOS Biology

cschnell@plos.org